

# Aerosol and cloud top height information of Envisat MIPAS measurements

Sabine Griessbach[1], Lars Hoffmann[1], Reinhold Spang[2], Peggy Achtert[3], Marc von Hobe[2],
Nina Mateshvili[4], Rolf Müller[2], Martin Riese[2], Christian Rolf[2], Patric Seifert[5], and Jean-Paul Vernier[6]

[1]Jülich Supercomputing Centre (JSC), Forschungszentrum Jülich GmbH, Jülich, Germany
[2]Institute of Energy and Climate Research (IEK-7), Forschungszentrum Jülich GmbH, Jülich, Germany
[3]Department of Meteorology, Stockholm University, Stockholm, Sweden; now at Department of Meteorology, University of Reading, Reading, UK
[4]Belgian Institute for Space Aeronomie, Brussels, Belgium & Abastumani Astrophysical Observatory, Ilia State University, Tbilisi, Georgia
[5]Leibniz Institute for Tropospheric Research (TROPOS), Leipzig, Germany
[6]NASA Langley Research Center, Hampton, VA, USA

*Correspondence to:* Sabine Griessbach (s.griessbach@fz-juelich.de)

**Abstract.** Infrared limb emission instruments have a long history in measuring clouds and aerosol. In particular the Michelson Interferometer for Passive Atmospheric Sounding (MIPAS) instrument aboard ESA's Envisat provides 10 years of altitude resolved global measurements. Previous studies found systematic over- and underestimations of cloud top heights for cirrus and polar stratospheric clouds. To assess the cloud top height information and to characterize its uncertainty for the MIPAS instrument we performed simulations for ice clouds, volcanic ash, and sulfate aerosol. From the simulation results we found that in addition to the known effects of the field-of-view that can lead to a cloud top height overestimation, and broken cloud conditions that can lead to underestimation, also the cloud extinction plays an important role. While for optically thick clouds the possible cloud top height overestimation for MIPAS reaches up to 1.6 km due to the field-of-view, for optically thin clouds and aerosol the systematic underestimation reaches 5.1 km. For the detection sensitivity and the degree of underestimation of the MIPAS measurements also the cloud layer thickness plays a role. 1 km thick clouds are detectable down to extinctions of $5 \times 10^{-4}$ km$^{-1}$ and 6 km thick clouds are detectable down to extinctions of $1 \times 10^{-4}$ km$^{-1}$, where the largest underestimations of the cloud top height occur for the optically thinnest clouds with a vertical extent of 6 km. The relation between extinction coefficient, cloud top height estimate, and layer thickness is confirmed by a comparison of MIPAS cloud top heights of the volcanic sulfate aerosol from the Nabro eruption in 2011 with space- and ground-based lidar measurements and twilight measurements between June 2011 and February 2012. In the fresh to two months old plume, where the extinction was between $1 \times 10^{-4}$ and $7 \times 10^{-4}$ km$^{-1}$ and the layer thickness mostly below 4 km, we found for MIPAS an average underestimation of 1.1 km. In the aged plume with extinctions down to $5 \times 10^{-5}$ km$^{-1}$ and layer thicknesses of up to 9.5 km the underestimation was higher reaching 7.2 km. The dependency of the cloud top height over- or underestimation on the extinction coefficient can explain seemingly contradictory results of previous studies. In spite of the relatively large uncertainty range of the cloud top height, the comparison of the detection sensitivity towards sulfate aerosol between MIPAS and a suite of widely used UV/VIS



limb and IR nadir satellite aerosol measurements shows that MIPAS provides complementary information in terms of detection sensitivity.

# 1 Introduction

Stratospheric aerosol is known for its impact on radiation and climate (e.g. Crutzen, 2006; Tilmes et al., 2009; Solomon et al., 2011; Santer et al., 2014), atmospheric dynamics (e.g. Wegmann et al., 2014; Diallo et al., 2017) and chemistry (e.g Tilmes et al., 2008; Wegner et al., 2012; Solomon et al., 2016). Aerosol in the upper troposphere is also of relevance to the Earth's radiation budget (Ridley et al., 2014). The study by Ridley et al. (2014) shows that aerosol in the upper troposphere and lower stratosphere (UTLS) can contribute significantly to the Earth's radiation budget and that the altitude information is important to estimate the radiative impact. Moreover, the available global long-term observations of stratospheric aerosol have a measurement gap in the UTLS at mid- and high latitudes (Ridley et al., 2014). This measurement gap has two reasons. First, global long-term measurements are provided by UV/VIS limb instruments (e.g. solar occultation and solar scattering measurements) that do not measure during night and hence, miss the entire polar night. Second, as the discrimination between aerosol and thin ice is rather complicated in the UTLS region (Kent et al., 2003) many stratospheric aerosol products terminate at around 15 km and therefore miss part of the lower stratosphere (Ridley et al., 2014). Further, at mid- and high latitudes many air traffic corridors are located in the UTLS and in case of volcanic eruptions volcanic material can pose a severe danger to air traffic (Casadevall, 1994; Guffanti et al., 2010). Hence, continuous global altitude resolved aerosol measurements are desirable.

In particular, volcanic eruptions have a significant impact on the stratospheric aerosol load (Kremser et al., 2016). In the tropics even moderate eruptions with a Volcanic Explosivity Index (VEI) of 4 and smaller that only reach the tropopause region contribute to the stratospheric aerosol layer (Vernier et al., 2011). Aside from direct injections the transport pathways of aerosol into the tropical stratosphere are still discussed (Bourassa et al., 2012b; Vernier et al., 2013; Fromm et al., 2013; Fairlie et al., 2014; Fromm et al., 2014). Recently it was shown that also extra-tropical volcanic eruptions can contribute to the tropical stratospheric aerosol layer (Schmidt et al., 2010; Diallo et al., 2017; Wu et al., 2017; Günther et al., 2018) and the corresponding transport pathways were uncovered (Wu et al., 2017). However, in order to distinguish between eruptions with plumes restricted to the troposphere and eruptions with stratosphere reaching plumes, and to study the transport pathways in the UTLS region global measurements with reliable information on the injection heights and aerosol altitudes are required.

Global altitude resolved measurements of stratospheric and upper tropospheric aerosol are provided by space-based limb measurements and active lidar measurements, e.g. Cloud-Aerosol Lidar with Orthogonal Polarization (CALIOP, Vernier et al., 2011). The limb measurements comprise the solar occultation, e.g. Stratospheric Aerosol and Gas Experiments II, Halogen Occultation Experiment, Polar Ozone and Aerosol Measurement (SAGE, HALOE, POAM, respectively, Randall et al., 2001), the stellar occulation, e.g. Global Ozone Monitoring by Occultation of Stars (GOMOS, Robert et al., 2016), the solar scattering, e.g. Optical Spectrograph and InfraRed Imager System (OSIRIS, Bourassa et al., 2012a), and the infrared limb emission, e.g. Cryogenic Limb Array Etalon Spectrometer (CLAES, Bauman et al., 2003) and Envisat Michelson Interferometer for Passive Atmospheric Sounding (MIPAS, Griessbach et al., 2016; Günther et al., 2018), measurement techniques.



Table 1 summarizes the relevant measurement characteristics of selected instruments representing common measurement techniques. In spite of the near-global coverage most instruments have significant temporal and spatial measurement gaps that make their data more suited to study long term changes and to develop climatologies than following individual aerosol plumes and identifying distinct sources. E.g. GOMOS provides dedicated aerosol measurements during nighttime, but misses the daytime hemisphere including the polar summer region. CALIOP provides day- and nighttime aerosol measurements, yet the nighttime measurements have a significantly higher sensitivity towards aerosol than the daytime measurements. In contrast, OSIRIS measures only at daytime and SAGE-II measured during sunrise and sunset. Figures showing the differences in the latitudinal coverage and the sampling frequencies of typical solar and stellar occultation, solar scattering, and infrared limb emission measurements can be found in Sofieva et al. (2013, Fig. 2) and Robert et al. (2016, Fig. 2).

Infrared nadir measurements provide an excellent global coverage at day- and nighttime, but have a significantly lower sensitivity towards aerosol than limb measurements due to their measurement geometry. It has been shown that enhanced volcanic sulfate aerosol can also be detected with infrared nadir measurements (Ackerman, 1997; Clarisse et al., 2013). However, in general, the altitude retrieval competes with the concentration and radius retrieval (Clarisse et al., 2010) and e.g. dust aerosol altitudes retrieved from Atmospheric Infrared Sounder (AIRS) measurements indicate the altitude where half the aerosol optical depth (AOD) of the entire layer is reached (Peyridieu et al., 2010).

Lidar measurements (ground based, air-borne, or space based) are considered to provide the best altitude information. However, studies comparing e.g. SAGE II with lidar measurements smoothed the vertically higher resolved lidar data to SAGE II vertical resolution (e.g. Antuña et al., 2002) and extinction profiles of, e.g. GOMOS and OSIRIS, have been validated against SAGE II and SAGE III (e.g. Robert et al., 2016; Bourassa et al., 2012a, respectively). Studies assessing or validating the aerosol and cloud altitude information from limb remote sensing measurements are rare. Only Kent et al. (1997) investigated the accuracy and potential error sources of SAGE II cloud heights. Using two flights of airborne lidar cloud measurements they simulated the signal SAGE II would receive for these clouds. For the 41 simulated scenarios Kent et al. (1997) found an overestimation of the cloud top height in a single case and an underestimation by 1 to 5 km in 17 cases (39 %). The underestimation was attributed to patchy clouds, where the cloud was located along the ray path, but not at the tangent point. For CALIOP a very good agreement of cloud and aerosol altitudes was found compared with in situ balloon and lidar measurements, and ground based lidar measurements (Vernier et al., 2009; Mioche et al., 2010; Kim et al., 2011). Compared with ground based lidar measurements Kim et al. (2008) found that the CALIOP cloud top and base heights generally agree within 0.1 km.

Due to MIPAS large field-of-view smoothing the signal over an altitude range of about 3 km, several studies investigated the viability of MIPAS cloud top heights. For polar stratospheric clouds (PSCs) Höpfner et al. (2009) found that MIPAS systematically underestimates the cloud top height by about 0.3 (at 15 km) and up to 2.6 km (at 27 km) compared to CALIOP. The comparison shows a clear altitude dependence and the large underestimation at high altitudes was attributed to broken cloud conditions. Castelli et al. (2011) investigated the effect of broken cloud conditions on MIPAS cloud top heights and confirmed the underestimation effect for 1D cloud detection methods, but showed that a 2D retrieval can lead to significant improvements if the cloud was sampled by consecutive MIPAS profile scans. For cirrus clouds Spang et al. (2012) quantified MIPAS' underestimation up to 2.5 km compared to SAGE II. In contrast, MIPAS was found to overestimate cirrus cloud



top heights by usually less than 2 km compared to Geoscience Laser Altimeter System (GLAS) measurements during the
Space Shuttle Mission (Spang et al., 2012). Also Sembhi et al. (2012) reported an overestimation of about 1 km compared to
CALIOP and of about 0.75 km compared to HIRDLS in a comparison of 3-monthly means of aerosol and cloud top heights.
This overestimation was attributed to MIPAS' field-of-view, where optically dense clouds below the tangent point, but within
the field-of-view, affect the measured radiances. Based on simulations of idealized clouds Hurley et al. (2009) proposed an
operational retrieval framework that is supposed to retrieve cloud top heights with an error below 0.5 km.

As the MIPAS measurements have the potential to fill gaps in the measurements of UTLS aerosol and clouds, the goal of this
study is to asses the cloud top altitudes derived from the MIPAS index cloud detection methods (Spang et al., 2004; Sembhi
et al., 2012; Griessbach et al., 2016), to reconcile the contradictory results of previous MIPAS cloud top height studies, and to
add a comparison for sulfate aerosol, which has not been investigated explicitly before. This study is based on MIPAS radiance
simulations for aerosol and ice clouds and a comparison between MIPAS and CALIPSO, ground based lidar, and twilight
measurements of the Nabro volcanic sulfate aerosol between June 2011 and January 2012. The instruments and aerosol data
products for MIPAS, CALIOP, the ground based lidars, and the twilight measurements are introduced in Sect.2. In Sect. 3
the simulated aerosol and cloud scenarios are described and the derived top heights are analyzed and assessed. The dedicated
comparison of the top heights for the Nabro sulfate aerosol is presented in Sect. 4. A discussion of the simulation results and
the comparison of the cloud top heights for the volcanic sulfate aerosol follows in Sect. 5. The conclusions are given in Sect. 6.

## 2 Instruments and data sets

### 2.1 MIPAS

MIPAS was an infrared limb sounder that was mounted on ESA's Envisat and operated from June 2002 to April 2012. It mea-
sured up to 1344 vertical profiles per day (14.4 orbits with 96 profiles) at day- and nighttime covering the latitudes between
$89.4°N - 87.3°S$. The high-resolution emission spectra in the thermal infrared ($685 - 2410\,cm^{-1}$) comprise altitudes between
6 to 68 km from July 2002 to March 2004 and 7 to 72 km from January 2005 to April 2012 in MIPAS' nominal mode (Fis-
cher et al., 2008). The nominal mode was changed due to a malfunction of the instrument in 2004. From 2002 to 2004 the
vertical sampling was 3 km and in 2005 the vertical sampling was decreased to 1.5 km below 21 km altitude. In addition, the
formerly homogeneous measurement geometry was changed, so that the lower measurement heights approximately follow the
tropopause, i.e. in the tropics MIPAS sampled down to 10 km and in the polar regions down to 7 km. The MIPAS engineer-
ing tangent altitudes, which are corrected for refraction, have a negative offset of about 300 to 500 m compared to retrieved
altitudes in the UTLS height range (Kleinert et al., 2018). The vertical widths of the field-of-view of MIPAS is about 3 km.
For retrieval of MIPAS measurements, the field-of-view is often assumed to be trapezoidal with edge lengths of 2.8 and 4 km
(Ridolfi et al., 2000; Hurley et al., 2009). The horizontal field-of-view, which is perpendicular to the line-of-sight, is 30 km
(Fischer et al., 2008). MIPAS measured at day- and nighttime with a mean local solar time of around 22:00 for the ascending
node and 10:00 for the descending node.





To derive aerosol we used the MIPAS band A ($685 - 970\,\text{cm}^{-1}$) and band B ($1215 - 1500\,\text{cm}^{-1}$) level 1b dataset processed with the instrument processing facility (IPF) processor version 7.11 that provides calibrated and geolocated radiances. A method to detect aerosol in the UTLS and filter out ice clouds has been introduced by Griessbach et al. (2016). This aerosol detection methods relies on two steps. First, for improved aerosol and cloud detection the aerosol cloud index (ACI) with

125 values smaller than 7 is used. The ACI is the maximum of the cloud index and the aerosol index that are the color ratios between the $CO_2$ band at $[788.25, 796.25]\,\text{cm}^{-1}$ and two atmospheric window regions at $[832.31, 834.37]\,\text{cm}^{-1}$ and $[960.00, 961.00]\,\text{cm}^{-1}$, respectively. In the second step the discrimination between aerosol and ice clouds is performed using threshold functions for brightness temperature difference correlations. Each spectrum of a profile is evaluated with this two step method and is identified as either clear air, aerosol, or ice cloud with the corresponding tangent height as altitude information. In order

to ensure that the MIPAS aerosol measurements used in our study only comprise sulfate aerosol, first the aerosol and cloud detection was performed, the ice clouds were filtered out as described above, and finally the volcanic ash detection method (Griessbach et al., 2014), which was also found to be sensitive to mineral dust and wild fire aerosol, was applied to filter out other non-sulfate UTLS aerosol types.

## 2.2   Space-based lidar CALIOP

CALIOP is an active lidar instrument on-board the Cloud-Aerosol Lidar and Infrared Pathfinder Satellite Observation (CALIPSO) that flies in the A-Train constellation since 2006 (Winker et al., 2009). The satellite is in a sun-synchronous polar orbit with the local equator-crossing time of the ascending node at about 13:30 and of the descending node at about 1:30 that allows for measurements at day- and nighttime between $82\,°$N and $82\,°$S (Winker et al., 2009). CALIOP is a near-nadir viewing two wavelength polarization-sensitive lidar (Winker et al., 2009). The best signal-to-noise ratio is provided by the nighttime

measurements at $532\,\text{nm}$ (Vernier et al., 2009).

Focusing on aerosol measurements in this study, a dedicated aerosol product providing the highest sensitivity is used: After applying a cloud mask that is based on the ratio between the perpendicular and total backscatter signal at $532\,\text{nm}$, CALIOP nighttime profiles were averaged horizontally over $1\,°$ latitude and vertically over $200\,\text{m}$ (Vernier et al., 2009). Each day reflections of 1.7 million laser shots are acquired (Winker et al., 2009). Assuming $111\,\text{km}$ averaging, $40,009\,\text{km}$ circumference of

145 the Earth from pole to pole, and 14.55 orbits per day this results in about 2800 independent nighttime profiles. The optimized aerosol measurements cover the winter hemisphere up to $82\,°$ and the summer hemisphere up to $50\,°$ and have a further gap due to the South Atlantic Anomaly (Vernier et al., 2009). For the comparison of the measurements of the Nabro aerosol in boreal summer we used data between 0 and $50\,°$N and $12 - 20\,\text{km}$. The Nabro aerosol was visible in individual CALIOP profiles between 15 June to 11 August 2011. To convert the backscatter signal to extinction coefficient we used a constant lidar ratio

(aerosol extinction to backscatter ratio) of $50\,\text{sr}$ as Sawamura et al. (2012) mostly used this ratio and reported one measured lidar ratio of $48\,\text{sr}$ at $532\,\text{nm}$ for the Nabro sulfate aerosol.





## 2.3 Ground-based lidars

### 2.3.1 Leipzig, Germany

In Leipzig (51.34° N 12.37° E), regular lidar measurements are performed with the Multiwavelength Atmospheric Raman
lidar for Temperature, Humidity, and Aerosol profiling (MARTHA) (Mattis et al., 2002) in the framework of the European
Aerosol Research Lidar Network (EARLINET, Pappalardo et al., 2014) three times each week, i.e., Monday afternoon, and
Monday and Thursday after sunset, when weather conditions allowed (i.e., absence of precipitation and low clouds). The light
source is a Nd:YAG laser that generates laser pulses at 355, 532, and 1064 nm wavelength with a repetition rate of 30 Hz. The
backscattered radiation is collected by a 0.8 m Cassegrain telescope. The aerosol backscatter coefficient can be determined
at 355, 532, and 1064 nm and the aerosol extinction coefficient at 355 and 532 nm is derived by means of the Raman-lidar
method (Ansmann et al., 1992) (not available for all layers). The system detects the component of light cross-polarized to the
plane of polarization of the outgoing beam at 532 nm. MARTHA observations have been used previously for the statistical
characterization of free-tropospheric aerosol layers (Mattis et al., 2008), volcanic aerosol layers (Mattis et al., 2010) and for
the characterization of polar stratospheric clouds (Jumelet et al., 2008).

The aerosol signal was derived from backscatter signals under cloud free conditions only, by averaging the data between
27 minutes and 6 hours (in average 2:09 h). From the averaged signal the layer top and bottom altitude were derived from
the gradient of the averaged signal profile following Mattis et al. (2008). After the cloud boundary determination the optical
depth was calculated at 532 nm wavelength from the Raman-derived aerosol backscatter coefficient for layers observed during
nighttime (Ansmann et al., 1992). For each nighttime layer we also computed the average extinction at 532 nm. The conver-
sion from backscatter coefficient to extinction coefficient requires the assumption of a backscatter-to-extinction (lidar) ratio.
Although for Leipzig measurements of stratospheric aerosol usually a lidar ratio of 30 sr is used, which was derived from
measurements within the first 2 years after the eruption of the Pinatubo volcano (Mattis et al., 2008), here we used a lidar ratio
of 50 sr to make the average layer extinctions comparable to CALIOP. The choice of a higher lidar ratio is further supported by
Sawamura et al. (2012), who derived a lidar ratio of 48 sr from measurements of Nabro aerosol, and Prata et al. (2017), who
derived lidar ratios around 60±15 sr in the first two weeks after the sulfur rich eruptions of Kasatochi and Sarychev. The error
in the layer boundary heights and particle backscatter coefficients is approximately 300 m and 20 %, respectively, due to the
low signal-to-noise ratio and small signal gradient produced by the sulfate layers of the Nabro volcano. The Nabro aerosol was
measured over Leipzig between 5 July 2011 and 10 February 2012 on 47 days (59 lidar profiles). These profiles were checked
for other aerosol sources, such as Grimsvötn sulfate aerosol (Tesche et al., 2011) and wild fires, that were removed from the
data for the altitude comparison.

### 2.3.2 Jülich, Germany

A ground based cloud lidar was operated by the research center Jülich located in the western part of Germany (50.92°N,
6.36°E, 91 m a.s.l.). The lidar system is a commercial lidar instrument (Leosphere, ALS 450) that operates at a wavelength
of 355 nm with a pulse energy of 16 mJ, a pulse duration of 4 ns, and a frequency of 20 Hz (Rolf et al., 2012). The parallel





and perpendicular polarized backscattered light is measured by two detectors to determine the depolarization that provides information on the sphericity of the scattering particles. The receiver telescope has a diameter of 15 cm with a full-angle field-of-view of around 1.5 mrad. The covered altitude range is from 0.5 to 19 km with a vertical resolution of around 30 m depending on atmospheric conditions.

The lidar is usually operated in times where potentially cirrus clouds are present. Thus, the operation is irregular in time and only complete cloud free sequences where selected for this study. The Nabro aerosol was measured over Jülich between 15 July and 12 December 2011 on 7 days (11 lidar profiles). The backscatter profiles used in this study were vertically smoothed with a window length of 360 m to increase the signal to noise ratio above 12 km. Sawamura et al. (2012) reported lidar ratios of 45 and 55±1 sr at 355 nm for the Nabro aerosol. As in both cases the lidar ratio at 355 nm was 7 sr larger than at 532 nm for the same sample, we used a lidar ratio of 55 sr. The aerosol layer top and bottom altitudes were derived based on a comparison of the signal to the signal to noise ratio. Additionally, it was checked that the thermal tropopause derived from ERA-interim reanalyses is below the Nabro aerosol layer.

### 2.3.3 Esrange, Sweden

The Department of Meteorology of the Stockholm University operates a lidar at Esrange (67.89°N, 21.10°E) near the Swedish city of Kiruna (Blum and Fricke, 2005; Achtert et al., 2013). The Esrange lidar uses a pulsed Nd:YAG solid-state laser as light source. A detection range gate of $1\,\mu$s results in a vertical resolution of 150 m. The backscattered light at 532 nm is detected in two orthogonal planes of polarization. Measurements of backscattered light in the parallel and perpendicular channels were used to gain information on the shape of the scattering particles, since perpendicularly polarized signals only occur when scatterer are of non-spherical shape. The molecular fraction of the received signal was determined from the vibrational Raman signal.

Measurements with the Esrange lidar were conducted on campaign base. The lidar was in operation between the 4 and the 25 January 2012, measuring on 6, 9, 13, 14, 19, 20, and 25 January. Within this study the parallel and perpendicular backscatter ratios, the particle backscatter coefficient, and the linear particle depolarization ratio were used. The extinction coefficients presented here have been obtained using the Esrange lidar's rotational Raman channels (Achtert et al., 2013), that do not require any lidar ratio. Measurements with these channels have also been used to derive the temperature profile from which the tropopause height has been inferred.

### 2.4 Twilight measurements

Spectral photometric measurements of the twilight sky brightness are performed in Tbilisi, Georgia (41.72° N, 44.78° E), on a regular basis every clear day at solar zenith angle range $90° - 96°$ (Mateshvili et al., 2005). From 2009 a CCD camera with a grating spectrograph is used to record a time series of spectral images between 685 -– 800 nm. The images are averaged over the wavelength interval 777.5 to 782.5 nm and the corresponding acquisition time moments are converted to solar zenith angles. The thus obtained dependencies of the monochromatic brightness of the twilight sky on the solar zenith angles are





used to retrieve aerosol extinction vertical profiles. The aerosol extinction profiles at 780 nm and the corresponding errors were retrieved in the UTLS (Mateshvili et al., 2013).

The post-Nabro aerosol profiles were acquired 11 times between 14 July and 03 August 2011 from the observational site
Tbilisi. In this study we used the derived extinction profiles at 780 nm ($12820.5\,\mathrm{cm}^{-1}$) and the corresponding errors that are given on a 1 km grid. The actual vertical resolution of the retrieved aerosol extinction profiles were estimated from the half-widths of the maximums of the averaging kernels at the respective altitudes up to $1.5 - 2\,\mathrm{km}$ for the Nabro aerosol layer (Mateshvili et al., 2013).

These twilight data is the only ground-based data set that has an overlap with the CALIOP measurements. As it relies on
an entirely different technique than the lidar measurements it further allows for checking the consistency between the various aerosol layer top height definitions.

## 3 Radiative transfer simulations of aerosol and cloud layers

### 3.1 Simulation setup

The radiative transfer model used for the MIPAS simulations is the *J*ülich *R*apid *S*pectral *S*imulation *C*ode (JURASSIC)
(Hoffmann et al., 2008). The fast radiative transfer and retrieval model was used to calculate spectrally averaged radiances on MIPAS' spectral grid based on pre-calculated emissivity look-up tables and the emissivity growth approximation (EGA) (Weinreb and Neuendorffer, 1973; Gordley and Russell, 1981). JURASSIC was extended to account for single scattering on aerosol and cloud particles where the optical properties were calculated using Mie theory (Griessbach et al., 2013).

For the simulation of IR limb emission spectra in the presence of clouds and aerosol we followed the setup described in
Griessbach et al. (2014) with some modifications. We placed three 1 km thick homogeneous sulfate aerosol, ice cloud, and volcanic ash layers at $9 - 10$, $13 - 14$, $17 - 18$ km, and one 6 km thick sulfate aerosol layer at $10 - 16$ km altitude for the four representative atmospheric states of a polar winter, polar summer, mid-latitude, and equatorial atmosphere (Remedios et al., 2007). Although MIPAS samples the UTLS at 1.5 km in the vertical, the pencil beam simulations were performed on a 0.5 km grid and interpolated to a 0.1 km grid after applying MIPAS' trapezoidal field-of-view, in order to account for the fact that real
clouds are located at any vertical distance to the sampling altitudes. The simulations cover the tangent height range between 8.5 and 20 km.

For the particle shape we assumed spherical particles, which is realistic for the liquid sulfate aerosol and an approximation for ice and volcanic ash. The IR refractive indices were taken from Hummel et al. (1988) for sulfate aerosol (75 % sulfuric acid solution), Warren and Brandt (2008) for ice, and Pollack et al. (1973) for basalt as a representative for volcanic ash. Each
particle type has its individual realistic particle size and extinction coefficient ranges as reported by in-situ measurements (see Griessbach et al., 2014, for a discussion of the ranges). For sulfate aerosol we considered mode radii between 0.01 and 1.5 $\mu$m of a lognormal size distribution with a width of 1.6 (which corresponds to effective radii between 0.02 and 2.6 $\mu$m) and selected the particle number concentration such that it resulted in extinction coefficients between $1 \times 10^{-4}$ and $1 \times 10^{-2}\,\mathrm{km}^{-1}$ in half an order of magnitude steps at $950\,\mathrm{cm}^{-1}$. For ice clouds the mode radius range was 0.3 to 96 $\mu$m and the extinction coefficient





range $1 \times 10^{-3}$ to $1\,\mathrm{km}^{-1}$, and for volcanic ash the mode radii ranged from 0.1 to $5\,\mu\mathrm{m}$ and the extinction coefficients included $1 \times 10^{-3}$ to $5 \times 10^{-1}\,\mathrm{km}^{-1}$.

## 3.2  Results: MIPAS detection sensitivity and top height uncertainties

To detect aerosol and clouds with MIPAS we used the aerosol-cloud-index (ACI) and considered the cloud top as the first tangent altitude where the ACI falls below 7 (Griessbach et al., 2016). In the following the term "cloud top height" refers to the
top altitude of ice cloud, ash, and sulfate aerosol layers. A comparable sensitivity to the cloud index (CI) method (Spang et al., 2005) using altitude and latitude dependent thresholds by Sembhi et al. (2012) was reported in Griessbach et al. (2016) for this method. We found that the 1 km thick aerosol layers are detectable down to an extinction coefficient of $5 \times 10^{-4}\,\mathrm{km}^{-1}$, which is in agreement with previous results by Griessbach et al. (2016). In addition, we found that the 6 km thick aerosol layer is in some (but not all) cases detectable down to the extinction coefficient of $1 \times 10^{-4}\,\mathrm{km}^{-1}$. This is because the vertically larger
layer fills a larger volume of the field-of-view. Figure 1a shows that the field-of-view volume that is filled with aerosol/cloud at the detected layer top height is a function of aerosol/cloud extinction coefficient. Further, Fig. 1a indicates that the field-of-view volume at the detected cloud top does not depend on the particle type. This justifies our approach of combining ice cloud, ash and sulfate aerosol simulations for the characterization of the cloud top uncertainty. The variability is rather a function of the tangent altitude.

The dependency of the detection sensitivity on the extinction coefficient and the field-of-view volume filled with aerosol/cloud also has implications for the detected cloud top height. The fine grid simulation results show that the cloud top height mainly depends on the extinction coefficient, some effect can be seen for the background atmosphere, and nearly no effect can be seen for the different particle types (Fig. 1b). Transferring the fine grid simulation results to the coarser MIPAS vertical sampling, the resulting maximal altitude ranges around the "real" cloud top height are given in Table 2. The potential uppermost
detected top height on the MIPAS sampling grid is the same as on the fine grid. The lowermost top height results from the following consideration: If the cloud top is detected at 18 km on the 100 m simulation grid and the MIPAS tangent heights are at 18.1 and 16.6 km (1.5 km vertical sampling), the measured MIPAS cloud top height would be 16.6 km. While the top height of thick clouds can be overestimated by up to 1.6 km due to the field-of-view, the cloud top of clouds with an extinction of $1 \times 10^{-3}\,\mathrm{km}^{-1}$ and smaller can be underestimated by up to 5.1 km (for the 6 km thick layer) due to the field-of-view and
extinction effect (Table 2). Depending on the position of the cloud relative to the tangent height the detected top height may fall anywhere in-between the ranges given in Table 2.

In practice, when analyzing MIPAS measurements the extinction coefficient of any cloud or aerosol layer is unknown unless an extinction retrieval is available. Hence, one cannot judge if the cloud top height is over- or underestimated. The simulated ACI at the cloud top is always close to 7, however, analyzing the entire profile provides more information. The cloud layers
with an extinction of $1 \times 10^{-2}\,\mathrm{km}^{-1}$ or smaller result in profiles with a pronounced local ACI minimum (e.g. see the measured MIPAS profile in Fig. 3), whereas for thicker clouds the ACI profiles run into saturation with ACI values around 2 (Fig. 1c). From Fig. 1d, showing the relation between the minimum ACI of a profile and the height difference, we deduce that for saturated ACI profiles the derived cloud top height overestimates the real cloud top height in most cases. For profiles with





a local ACI minimum the cloud top height can be over- or underestimated and the minimum ACI can be used as a rough

indicator for the likelihood of over- or underestimation (Fig. 1d). For minimum ACIs smaller than 3 there is a strong tendency

towards overestimation and the possible underestimation is less than $-1.5\,\text{km}$. For minimum ACI values larger than 5.5 there

is a strong tendency towards underestimation and the overestimation is less than $0.6\,\text{km}$. In general, the smaller the minimum

ACI the more likely is an overestimation and the larger the minimum ACI the more likely is an underestimation of the cloud

top height.

## 4   Comparison of measured Nabro sulfate aerosol cloud top heights

### 4.1   Nabro eruption

For the comparison with other measurements we focus on the eruption of the Eritrean Nabro volcano that started on 12 June

2011 around 20:32 UTC (Goitom et al., 2015). With an $SO_2$ emission of about $4.5\,\text{MT}$ within fifteen days (Theys et al., 2013)

this eruption released the largest amount of $SO_2$ during the MIPAS measurement period between 2002 to 2012. This eruption

is known to have injected little to no ash particles (Theys et al., 2013). The injection reached the UTLS region (Sawamura

et al., 2012), where exact knowledge of the altitude is crucial for scientific conclusions (Bourassa et al., 2012b; Vernier et al.,

2013; Fromm et al., 2014).

### 4.2   Method

The Nabro sulfate aerosol in the UTLS was measured by inherently different measurement techniques that have individual

sensitivities with respect to the aerosol layer extinction coefficient, top height, and thickness. Hence, we compared the various

definitions of aerosol layer top height and the corresponding aerosol detection sensitivities of the instruments used in this study

individually. For the MIPAS measurements we used ACI$\leq$7, the ice, and ash cloud filter methods (Sect. 2.1) to identify the

cloud top height of the Nabro aerosol layers. From the simulation study we already derived that aerosol layers with extinctions

down to $1 \times 10^{-4}\,\text{km}^{-1}$ are detectable and that the detection height and the corresponding uncertainty range strongly depend

on the layer extinction (Sect. 3.2).

   To make the MIPAS measurements with its detection limit compareable to the lidar and twilight measurements we esti-

mated the scaling factors between the wavelengths of the lidar, and twilight measurements and the wavelength the MIPAS

simulation results are valid for ($10.5\mu$m) (Fig. 2). As the extinction coefficient of sulfate aerosol has a strong dependency on

the wavelength (wavenumber) and particle size, we used five measured sulfate aerosol size distributions (PSD) after the Mt

Pinatubo (Deshler et al., 1992b, a, 1993, 4 PSDs) and Nabro (Bourassa et al., 2012b, 1 PSD) eruptions and calculated the

corresponding extinction coefficient spectra from 0.2 to $16\,\mu$m using Mie calculations and the refractive indices of Hummel

et al. (1988) for 75% sulfuric acid solution. Further, we fitted 3 representative mono-modal log-normal PSDs to the minimum

($\mu = 0.07 \equiv r_{\text{eff}} = 0.12\mu$m), maximum ($\mu = 0.23 \equiv r_{\text{eff}} = 0.4\mu$m), and middle extinction ($\mu = 0.14 \equiv r_{\text{eff}} = 0.24\mu$m) spectra

assuming a distribution width of 1.6 (Fig. 2). The spectra are normalized to $948.5\,\text{cm}^{-1}$ (dashed line) as this is the wavenumber





used to set the extinction coefficients for our MIPAS simulations. The scaling factors from $948.5\,\mathrm{cm}^{-1}$ ($10.5\,\mu\mathrm{m}$) to the wavelengths used for the lidar, twilight measurements, and selected satellite instruments introduced in Sect. 1 are given in Table 3 as a range (minimum to maximum) for the measured PSDs and fitted PSDs individually.

The individual cloud top height definitions of the CALIOP and ground based measurements used for the comparison are presented in the following Sections 4.3 – 4.7 individually as well as the chosen match criteria and the results.

### 4.3  CALIOP measurements

Most global comparisons between MIPAS and CALIOP were performed on a statistical basis using temporal and regional mean cloud top height values, because ice clouds are highly variable in space and time. Especially in the tropics the match times (about 3:30 h difference in local equator crossing time) and match distances between MIPAS and CALIPSO are rather large in terms of cloud extent scales and formation time scales (e.g. Hurley et al., 2011; Sembhi et al., 2012; Spang et al.,
2012, 2015). For PSCs, which persist on longer time scales and have a larger horizontal extent than convective clouds, Höpfner et al. (2009) compared individual profiles of MIPAS and CALIOP measurements. Volcanic sulfate aerosol we consider also sufficiently temporally persistent to allow for a comparison of individual profiles (Kent et al., 1997), but still keep in mind that very fresh plumes can be very localized. We set the temporal match time for the comparison of MIPAS and CALIOP aerosol detections to 6 h, as Höpfner et al. (2009). As a match radius we selected 500 km, which is the along track distance between
two MIPAS measurements. Given the fact that the longitudinal distance between two subsequent MIPAS (and CALIOP) orbits increases from about 500 km at $80°$ N to 1800 km at $50°$ N, and to 2800 km at the equator the choice of 500 km allows for a sufficient number of matches for statistics between $50°$ N and the equator. The choice of a 500 km match radius for an aerosol measurement comparison is further supported by the match criteria discussed by Antuña et al. (2002) for SAGE II ranging from $\pm 1-5$ degree in latitude (about $111-555$ km) and $\pm 1-25$ degree in longitude (about $111-2775$ km at the equator and
$72-1800$ km at $50°$ N ) and their finding that for aged plumes the results did not depend on the match criteria. As CALIOP has a much higher along track sampling rate than MIPAS (Fig. 5) we calculated the mean extinction profile of all CALIOP profiles within the match radius. In total we found 1190 MIPAS aerosol profiles with a matching CALIOP profile within the first 8 weeks after the Nabro eruption (12 June to 11 August 2011).

For comparing the cloud top heights the definition of the cloud top is crucial. While for MIPAS the aerosol detections are
defined by the ACI threshold and the cloud top height is the highest tangent altitude where the ACI falls below 7, which is sensitive to extinctions down to $1 \times 10^{-4}\,\mathrm{km}^{-1}$ (at $10.5\,\mu\mathrm{m}$), for the CALIOP extinction profiles we first used a detection sensitivity threshold and second used the extinction gradient to derive the cloud top height.

#### 4.3.1  Results for extinction threshold method

CALIOPs nominal extinction threshold (at 532 nm) is $5 \times 10^{-3}\,\mathrm{km}^{-1}$ at 18 km and $1 \times 10^{-2}\,\mathrm{km}^{-1}$ at 12 km altitude for night-
time measurements averaged over 80 km horizontally and 60 m vertically and assuming a lidar ratio of $50\,\mathrm{sr}^{-1}$ (Winker et al., 2009). For the aerosol product used in this study the CALIOP data were averaged over 111 km horizontally and 200 m vertically. Hence, the detection limit should be even lower. Fig. 3 shows single CALIOP profiles and the corresponding averaged





profile for a match with MIPAS. A match may contain between 1 and 9 single CALIOP profiles depending on where the CALIPSO track crosses the match radius. For averaging, only profiles were used where the maximum extinction exceeds

the detection sensitivity, i.e. clear air profiles were excluded from averaging. Around 15 km a clear aerosol signal is visible, whereas at altitudes above about 17 km and below about 14 km the signals get rather noisy. For the averaged profile the noisy signal is below $2\times10^{-3}\,\text{km}^{-1}$ and for the single profiles it is below $3\times10^{-3}\,\text{km}^{-1}$. This is a further indication that the detection limit of the dedicated CALIOP aerosol product is lower.

In order to find the CALIOP detection threshold that provides a comparable sensitivity towards sulfate aerosol to MI-

PAS, we first scaled the MIPAS detection limit of $1\times10^{-4}\,\text{km}^{-1}$ to the CALIOP wavelength (532 nm) yielding about $2.5-4.5\times10^{-3}\,\text{km}^{-1}$ (at 18 km) depending on the particle size distribution (see Table 3). This is lower than the detection limit for the standard CALIOP aerosol product of $5\times10^{-3}\,\text{km}^{-1}$ at 18 km (Winker et al., 2009). However, the detection limit of the aerosol product used in this study was expected to be somewhat lower than the limit for the standard product due to more averaging. To verify that the aerosol product used here has a lower detection limit, we counted the number of matches with

CALIOP when moving the altitude variable detection limit (Figure 3, dashed line) to lower and larger values. The results are given in Fig. 4 with the extinction coefficient of the altitude variable detection limit at 18 km on the x-axis. In total there are 1190 MIPAS aerosol profiles with a CALIOP match. For extinction thresholds between $1-7\times10^{-4}\,\text{km}^{-1}$ the number of matches is constant at 1153. It decreases by 1 at $8\times10^{-4}\,\text{km}^{-1}$, by 10 at $2\times10^{-3}\,\text{km}^{-1}$ and by 70 at $4\times10^{-3}\,\text{km}$ (Fig. 4). From this decrease we deduce that the detection sensitivities of MIPAS and the dedicated CALIOP aerosol product are comparable

for extinction thresholds ranging from about $2-4\times10^{-3}\,\text{km}^{-1}$ (at 532 nm). In the following the results for the altitude variable extinction threshold that is $3\times10^{-3}\,\text{km}^{-1}$ at 18 km are shown. That this choice is appropriate can be seen in Fig. 5 showing the top heights of MIPAS and CALIOP Nabro aerosol measurements and giving an impression of the data coverage of both measurements within 24 h.

The result of the comparison of the individual MIPAS and CALIOP aerosol profiles is shown in Fig. 6. MIPAS underesti-

mated the cloud top height of the Nabro sulfate aerosol by 1.08 km in median and 1.15 km in mean. In 95 % of all matches the cloud top height was underestimated and in 5 % it was overestimated. Individual differences reached up to +1.1 km and −5.9 km. In only three cases the overestimation was more than 1 km and in only two cases the underestimation was more than 5 km. This result is in line with the altitude uncertainty range given by the simulation results in Sect. 3, where the overestimation reached up to 1.6 km and the underestimation was up to 5.1 km. In 2.4 % of the matches the underestimation was more than

3 km. The analyses of the correlations between extinction coefficient and altitude difference (Fig. 7a), extinction coefficient and minimum ACI (Fig. 7b), and minimum ACI and altitude difference (Fig. 7c) show that the comparison results mainly fall within the ranges predicted by the simulations (compare with Fig. 1b, d, and c respectively). Outliers we attributed to potentially bad matches and in case of the smallest simulated extinction to the rather small set of simulated scenarios. The correlations also show that the Nabro aerosol was thin and not optically thick to MIPAS with ACI values larger than 2.5 (Fig. 7c) and extinctions

between $3\times10^{-3}-5\times10^{-2}\,\text{km}^{-1}$ in the VIS range being equivalent to $1\times10^{-4}-1.7\times10^{-3}\,\text{km}^{-1}$ in the IR (10.5 μm).

The sensitivity of the analysis to our selected extinction threshold was tested by performing the same analysis with extinction coefficient scaling factors of 20, 40, and 50. With a larger extinction threshold for CALIOP the number of matches decreased





and the average underestimation for MIPAS also decreased by about 200 m with each step, as the cloud top height is at a lower point in the aerosol layer. However, for all threshold values the top height was underestimated in $87 - 97\,\%$ of all cases

(Table 4).

### 4.3.2   Results for the gradient method

A common method to derive aerosol and cloud layer top and bottom altitudes from ground based lidar data is the gradient method (Mattis et al., 2008). Different from Mattis et al. (2008), we calculated the extinction gradient between two successive data points in each CALIOP profile and assigned the gradient to the altitude in-between (Fig. 3 middle). The maximum of

the gradient was used as the indicator for the cloud top height. The advantage of this method is that it is independent of any extinction threshold. The disadvantages, however, are that this method provides also results for noisy clear air profiles and that it misses thinner layers above a thick aerosol layer, because it provides only the overall maximum. To filter out clear air profiles we ran the analyses using extinction thresholds between $1{\times}10^{-4}$ and $5{\times}10^{-3}\,\mathrm{km}^{-1}$ as a pre-filter.

Compared to the CALIOP cloud top heights derived by using extinction coefficient thresholds the gradient methods provides

lower top heights in $60.4\,\%$ of all matches for a threshold of $2{\times}10^{-3}\,\mathrm{km}^{-1}$ and $13.6\,\%$ for $5{\times}10^{-3}\,\mathrm{km}^{-1}$ and higher top heights in $1.5\,\%$ of all matches for a threshold of $2{\times}10^{-3}\,\mathrm{km}^{-1}$ and $23.0\,\%$ for $5{\times}10^{-3}\,\mathrm{km}^{-1}$. The maximum agreement between both methods we found for $5{\times}10^{-3}\,\mathrm{km}^{-1}$ where $63.4\,\%$ of all cloud top heights agreed within 100 m. From visual inspection of the profiles we found that the gradient method provides lower cloud top heights than the extinction threshold method, because it misses thin aerosol layers above thicker layers and for smaller extinction thresholds the cloud top height moves up. For larger

extinction thresholds thinner layers are also filtered out and hence, both methods show a better agreement.

As the cloud top height derived by the gradient method does not depend strongly on the extinction threshold chosen as a pre-filter, the median and mean differences between MIPAS and CALIOP cloud top height are relatively constant ranging from $-0.8$ to $-0.9\,\mathrm{km}$ for extinction coefficient thresholds below $6{\times}10^{-3}\,\mathrm{km}^{-1}$ (Table 4). This result is comparable with the result using an extinction coefficient threshold of $4{\times}10^{-3}\,\mathrm{km}^{-1}$, yet the underestimating fraction is 7 percentage points smaller for

the gradient method, although still high with $84\,\%$ (Table 4). Since we found that the gradient method, as we implemented it, misses thinner aerosol layers above thick layers, we used the results from the extinction threshold method with a threshold of $3{\times}10^{-3}\,\mathrm{km}^{-1}$ in the further course of this study.

### 4.4   Leipzig lidar measurements

For the comparison of the ground-based lidar measurements with the MIPAS profiles we used a match radius of 500 km, as

for CALIOP, but a larger match time of the lidar measurement period $\pm 24\,\mathrm{h}$ as in several studies comparing SAGE II and ground based lidar aerosol measurements (e.g. Antuña et al., 2002; Kulkarni and Ramachandran, 2015). In total we found MIPAS aerosol measurements matching to 32 lidar profiles that were measured on 26 different days between 18 July 2011 and 02 February 2012 (e.g. Fig. 8a, b). For 16 nighttime lidar profiles extinction coefficients were available. We excluded one match on 13 January 2012 from the comparison, because MIPAS observed PSCs at altitudes above 20 km within the match

radius. (Although PSCs over central Europe are rare they have been reported over Leipzig before (Jumelet et al., 2009) and





other measurements confirmed this finding, e.g. CALIOP browse images on 14 January 2012, ~12:45 UTC.) In several cases there was more than one MIPAS profile within the match criteria. In these cases the top heights were analysed for each MIPAS profile and the minimum, maximum, and mean top heights are given in Table 5 including the lidar aerosol layer top and bottom altitudes, and the nighttime extinction coefficient. All matches where an aerosol layer was visible in the MIPAS ACI profiles,
but the ACI did not reach the detection threshold, were excluded from the top height comparison (e.g. Fig. 8c, where the profiles in clear air are noisy and then fall together in the aerosol layer region).

In the comparison of the top heights we found that MIPAS underestimates the aerosol layer top height in all matches (Fig. 9a). The underestimation ranges from 0.9 km to 7.2 km and is 3.4 km in mean and 3.1 km in median. From the simulations we derived that these underestimations can be expected for extinction coefficients smaller than $1 \times 10^{-3}\,\mathrm{km}^{-1}$ at $10.5\,\mu\mathrm{m}$,
which is smaller than about $3 \times 10^{-2}\,\mathrm{km}^{-1}$ at 532 nm. The average layer extinction coefficients for the nighttime measurements range from $1.6 \times 10^{-3}$ to $4.1 \times 10^{-3}\,\mathrm{km}^{-1}$ assuming a lidar ratio of 50, as for the CALIOP measurements (Fig. 9b). These extinction coefficients are below or close to the lowest aerosol detection limit that we derived from the simulations for MIPAS. Our simulations indicate that aerosol layers with low extinctions are only detectable if they are thicker than 1 km and that a significant top height underestimation can be expected. The accumulated vertical extent of the aerosol layers above 7 km,
which is approximately the lowest MIPAS tangent altitude within the match radius around Leipzig, ranges from 1.9 to 9.6 km (Fig. 9c). The comparison of the aerosol layer top heights between MIPAS and the Leipzig lidar is consistent with the findings from the simulations. Between July to November 2011 we observe a decrease in extinction coefficient, an increase in layer thickness, and consequently an increase in top height difference.

The gradient method following Mattis et al. (2008) was used to derive the layer top height, because the Leipzig lidar has
a higher sensitivity towards aerosol than CALIOP and consequently the extinction threshold method would miss layers with low extinctions. The sensitivity on the method used to derive the top height we investigated with the three selected nighttime profiles in Fig. 8, since extinction coefficients were only available for the nighttime measurements. Using the extinction coefficient threshold of $3 \times 10^{-3}\,\mathrm{km}^{-1}$ the top heights became 0.7 km lower in the first case, 3.1 km lower in the second case, and 6.0 km lower in the third case (Table 6). Consequently, this leads to smaller top height differences compared to MIPAS,
$-0.4$ to $-0.7$ km in the first case and $-1.2$ to $-3.8$ km in the second case. The third case would have been excluded from the comparison, because the top layer is below the extinction threshold and the bottom layer top height is below the lowest MIPAS tangent altitude of about 7 km around Leipzig. Using the extinction threshold method would bring the altitude difference, mean extinction, and layer thickness closer to the bulk of the CALIOP measurements in Fig. 9.

The finding that the gradient method derives higher top heights for ground based lidar than the extinction threshold method is
in contrast to the finding for CALIOP, where the gradient method derived lower top heights. One contribution to this dicrepancy is that for CALIOP we only considered the maximum gradient of the entire profile, which is only correct under the assumption of a single layer, and hence, in some cases missed a thinner layer with a smaller gradient above, whereas for each Leipzig lidar profile a more sophisticated and visual analysis was performed that provided multiple layer structures. Another contribution to this discrepancy is the difference in the sensitivity of the ground-based and space-based lidar profiles. While for the CALIOP





profiles the average layer exinction was always larger than $3 \times 10^{-3}\,\mathrm{km}^{-1}$ to be clearly detectable, the average layer exinction of the Leipzig lidar profiles was mostly below $3 \times 10^{-3}\,\mathrm{km}^{-1}$.

In several cases there was more than one MIPAS profile within the match range (Fig. 8). Although the MIPAS cloud top height often is not the first tangent altitude within the aerosol layer (Fig. 8) we observe that the profiles are noisy above the aerosol layer, but get aligned with the first tangent altitude within the aerosol layer. Even if there is no MIPAS aerosol detection,
we observe a clear aerosol signal in the MIPAS ACI profiles that are aligned within the aerosol layer (Fig. 8c).

### 4.5   Jülich lidar measurements

The Jülich cloud lidar in Germany was operated on several days in July, August and December 2011. The match criteria were the same as for the Leipzig lidar: a match radius of $500\,\mathrm{km}$ and a match time of start of the measurement $- 24\,\mathrm{h}$ to end of the measurement $+ 24\,\mathrm{h}$. To achieve a good signal to noise ratio the lidar profiles were averaged over measurement periods of 3 to 4
hours. In total 10 lidar profiles were available and we found matches with MIPAS aerosol measurements for 7 lidar profiles that were measured on 6 different days (e.g. two examples in Fig. 10). The lidar cloud top and bottom altitudes, average extinction and the MIPAS minimum, maximum, and mean cloud top heights are given in Table 7 for each match. As already shown for the aged aerosol measured over Leipzig (Fig. 8c), we observe that the MIPAS ACI profiles are directed below the aerosol top, but the ACI threshold (7) is only crossed at altitudes close to the aerosol layer bottom altitude (Fig. 10b).
The comparison of the cloud top heights shows that the MIPAS top heights are always below the lidar top heights, by $-0.9$ to $-5.4\,\mathrm{km}$, but always within the aerosol layer (Fig. 9a). Between August to December 2011 there is an increase in the top height difference, which is in agreement with the results for the Leipzig station. The average extinction coefficients at $355\,\mathrm{nm}$ are given in Table 7. To make the extinctions comparable to the CALIOP and the Leipzig lidar measurements, we derived a scaling factor of 0.46 from the data in Fig. 2 for the particle size distribution measured after the Nabro eruption to scale
the $355\,\mathrm{nm}$ extinction coefficient to $532\,\mathrm{nm}$. Figure 9b shows that the scaled extinction coefficients between $2.4 \times 10^{-3}$ and $7.4 \times 10^{-3}\,\mathrm{km}^{-1}$ are within the range of the CALIOP and Leipzig measurements. The aerosol layer thickness measured in Jülich in August is significantly larger than in the CALIOP measurements (Fig. 9c). This can be attributed to the fact that the CALIOP aerosol data is only available south of $50°$ and down to $12\,\mathrm{km}$. The profiles often terminate already around $15\,\mathrm{km}$ due to ice clouds, whereas the Jülich profiles cover the extra-tropical stratosphere and the troposphere down to the ground.
A sensitivity test using an extinction coefficient threshold to identify the aerosol layer top height did not work, because the extinction profiles were very noisy in the stratosphere and often reach the threshold at the uppermost tangent altitude.

### 4.6   Esrange lidar measurements

The Esrange lidar in north Sweden was operated on 8 days between the 6th and the 25th of January 2012 in order to measure PSCs. We applied the same match criteria as for the Leipzig and Jülich lidars: a match radius of $500\,\mathrm{km}$ and a match time of
start of the measurement $- 24\,\mathrm{h}$ to end of the measurement $+ 24\,\mathrm{h}$. PSCs have a strong impact on MIPAS measurements and the signal of the aerosol layer is disturbed by the PSC signal (as seen in Fig 11a). Hence, we excluded all matching MIPAS





profiles that were affected by PSCs. Finally we found 3 lidar profiles (13, 19, 23) with matching unperturbed MIPAS profiles (e.g. Fig 11).

For the Esrange lidar only the parallel and perpendicular backscatter ratios were available. To determine the sulfate aerosol
layer we considered all measurements where the parallel backscatter signal was larger than 1.003. In a second step we filtered out PSCs above 17 km and ice clouds where the perpendicular backscatter signal is larger than the parallel backscatter signal due to solid particles (e.g. Fig 11a). We also applied the gradient method (Sect. 4.4)) to the parallel backscatter profiles. In those cases where a clear variation in the gradient was present the cloud top heights of both methods agree. However, due to the low particle concentrations in aerosol layers the gradient method did not provide results in all cases (Mattis et al., 2008).
For the three matches the lidar top and bottom altitudes, average extinction and the MIPAS minimum, maximum, and mean top heights are given in Table 8.

In the first match on 13 January 2012 (Fig. 11a) the lidar profile also shows solid PSC particles in the stratosphere and 3 out of 7 MIPAS profiles within the match radius also detected these PSCs. However, 4 profiles were not affected by the PSCs. For all matches, the aerosol layer top height detected by MIPAS is $1.5 - 2.8$ km below the top height derived from the lidar
measurement (lidar measurement altitude error $\pm 150$ m). The aerosol layer thickness reaches from about 3 to 7.5 km, which is within the wide range found by the Leipzig lidar (Fig. 9c). Fig. 11b shows a match between PSC free lidar and MIPAS profiles on 20 January 2012. Although the MIPAS ACI profiles do not cross the detection threshold the aerosol layer signal is clearly visible (noisy profiles above the aerosol layer and aligned profiles with a local ACI minimum within the layer). For two days it was possible to derive an average extinction coefficient from the 532 nm lidar data that is shown in Fig. 9b as a representative
for the Esrange data.

## 4.7   Tbilisi twilight measurements

The match criteria for the comparison of the twilight profiles with MIPAS profiles were the same as for the lidar measurements: 500 km match radius and a match time of 18 UTC$\pm$24 h (an approximate time of evening twilights in summer). All MIPAS profiles within the match range were compared individually to the twilight profiles. In addition, we applied the same match
criteria to CALIOP measurements and added the averaged lidar profile to the comparison (Fig 12a,c). For the 11 twilight profiles measured between 14 July and 03 August 2011 we found matches with MIPAS aerosol measurements for 10 profiles and for CALIOP aerosol measurements we found matches for 8 profiles (Table 9).

For each twilight profile the top and bottom altitudes of the Nabro aerosol layer was estimated at extinction maximum half width. The top and bottom altitude uncertainty ranges are given by the intersection of the extinction at maximum half width
with the $\pm$ extinction error profiles (Fig 12b). While for all twilight profiles aerosol layer top and bottom heights could be derived, the full set of uncertainty ranges could be derived only for four profiles. The twilight top and bottom heights, the corresponding uncertainty ranges, the average layer extinction coefficient, the CALIOP top and bottom heights, the average CALIOP layer extinction coefficient, and the MIPAS cloud top heights are given in Table 9.

The differences between the twilight and MIPAS cloud top heights range from $-5.2$ to $+0.3$ km (Fig. 9a, and Table 9). In
all cases but one the MIPAS cloud top height is below the twilight cloud top height. Although for this particular measurement



no error estimate is available for the twilight measurement, assuming an error of $\pm 1.2\,\mathrm{km}$ from the other profiles the MIPAS and twilight top heights agree within the error range. Compared to CALIOP, the MIPAS top heights are always lower or equal. Comparing the twilight to the CALIOP top heights, the twilight top heights are in the range of $+3.4$ and $-1.1\,\mathrm{km}$ around the CALIOP top heights. In two cases this is above the twilight uncertainty. Although the twilight measurements have a

significantly coarser vertical resolution than the lidar measurements, the cloud top height differences to MIPAS are in the same range as for the lidar measurements (Fig. 9a). Also, the layer thickness derived from the twilight measurements is comparable to the layer thickness derived from the lidars (Fig. 9c). To compare the average layer extinction coefficient we scaled the twilight 780 nm extinction to the 532 nm lidar extinction using a scaling factor of 2.38 derived from Fig. 2 for the particle size distribution measured after the Nabro eruption. Except for 22 July 2011, where the signal was very weak and the extinction

coefficient was very low (out of plot range), the average twilight extinction agrees well with the lidar extinctions (Fig. 9b).

## 5 Discussion

### 5.1 Detection sensitivity

While Griessbach et al. (2016) showed that MIPAS measurements are sensitive to 1 km thick sulfate aerosol layers with extinction coefficients down to $5\times10^{-4}\,\mathrm{km}^{-1}$ the simulations in Sect. 3 showed that for 6 km thick layers the detection sensitivity

reaches down to $1\times10^{-4}\,\mathrm{km}^{-1}$. In the comparison with lidar and twilight measurements of the Nabro sulfate aerosol, MIPAS measurements were sensitive towards extinction coefficients of about $2\times10^{-3}\,\mathrm{km}^{-1}$ at 532 nm (Sect. 4, Fig. 9), which corresponds to about $6\times10^{-5}\,\mathrm{km}^{-1}$ at $10.5\,\mu\mathrm{m}$ (see Fig. 2). These extinctions are slightly higher than the lower aerosol and cloud detection limit of about $2\times10^{-5}\,\mathrm{km}^{-1}$ ($1\times10^{-5}\,\mathrm{km}^{-1}$ at $12\,\mu\mathrm{m}$ scaled to $10.5\,\mu\mathrm{m}$) found by Sembhi et al. (2012) for MIPAS.

Since the extinction coefficients for sulfate aerosol significantly differ between wavelengths in the VIS and IR range and

the detection sensitivities of the instruments introduced in Sect. 1 are given at their native wavelengths (Table 1), we used the scaling factors from Fig. 2 and Table 3 to compare the sensitivity range of MIPAS measurements towards sulfate aerosol with established instruments for aerosol detection. The sensitivity ranges given in Table 1, reaching from the lowest detectable extinction to the largest detectable extinction coefficient before becoming optically thick, were scaled to $10.5\,\mu\mathrm{m}$ ($950\,\mathrm{cm}^{-1}$) and are shown in Fig. 13. In brief, the solar occultation and scattering techniques provide the highest sensitivity towards sulfate

aerosol, but become optically thick at extinctions that are already reached by moderate volcanic eruptions (Fromm et al., 2014). The sensitivities of infrared limb emission measurements and active lidar are comparable and fill the gap in detection sensitivity between solar occultation/scattering and infrared nadir measurements.

### 5.2 Top height

The simulation results showed that for MIPAS the measured cloud top heights strongly depend on the layer extinction coeffi-

cient. For extinction coefficients of $1\times10^{-3}\,\mathrm{km}^{-1}$ (at $10.5\,\mu\mathrm{m}$) and smaller there is a strong tendency towards underestimation of the cloud top height, whereas for larger extinctions there is a strong tendency towards overestimation (Fig. 1). The extinc-





tion coefficients of the Nabro aerosol used for the comparison in this study range from $1.6 \times 10^{-3}$ to $4 \times 10^{-2}$ km$^{-1}$ at 532 nm (Fig. 9b), which corresponds to about $5.3 \times 10^{-5}$ and $1.3 \times 10^{-3}$ km$^{-1}$ at 10.5 $\mu$m. For this range of extinction coefficients the simulations (Sect. 3) predict a possible overestimation of up to about 1 km and possible underestimations of up to 5.1 km
(Fig. 1). The cloud top height differences deduced from the comparison of MIPAS to the lidar and twilight measurements between June and October 2011 (Fig. 9a) are in agreement with the results from the simulations, with an average underestimation of about 1 km by MIPAS, an overestimation in about 5 % of the cases of mostly below 1 km, and an underestimation of less than 5 km in most cases. From October 2011 on the altitude differences start to exceed $-5$ km. In most of these cases the extinction coefficient is smaller than the lowest simulated extinction coefficient in Fig. 1, but the measured layer thickness is larger than
6 km, which was the thickest layer assumed in the simulations. Hence, the comparisons of MIPAS measurements with lidar measurements indicate that the simulation results may be extrapolated towards thicker aerosol layers and smaller extinctions and consequently will lead to even larger top height underestimations. However, although the aerosol layers detected in the observations and discussed here can be considered sufficiently extended and homogeneous, we cannot rule out that some of the larger underestimations can also be attributed to broken cloud conditions. Considering the negative offset of about 0.3 to 0.5 km
in average of the MIPAS engineering tangent heights (Kleinert et al., 2018) there would be still an average underestimation of 0.7 to 0.5 km. Further, the majority of the compared profiles would still fall within the ranges predicted by the simulations.

Previous comparisons of MIPAS cloud top heights led to contradicory results. For cirrus clouds and aerosol Sembhi et al. (2012) found an overestimation of the cloud top height of up to 1 km in comparison to HIRDLS and CALIOP and Spang et al. (2012) found an overestimation of occasionally more than 2 km for MIPAS in comparison to the GLAS lidar. Spang et al.
(2012) also found an underestimation of up to 2.5 km for sub-visible cirrus (SVC) cloud top heights in comparison to SAGE II top heights. Further, Höpfner et al. (2009) reported an underestimation of up to 2.5 km in comparison to CALIOP for PSC top heights. These seemingly contradictory results can be reconciled when considering the characteristic extinction coefficients of the compared cloud data sets.

To make our results for the ACI detection method comparable to previous studies that used the CI for cloud detection, we
performed the entire analysis also for the CI using slightly modified altitude and latitude variable thresholds following Sembhi et al. (2012) (Griessbach et al., 2018). The CI shows a slightly higher sensitivity towards ice and a slightly lower sensitivity towards aerosol, but most importantly here, it systematically estimates the cloud top height 0.1 km higher than the ACI.

PSCs are optically relatively thin with 532 nm extinctions between about $1 \times 10^{-4}$ and $2 \times 10^{-2}$ km$^{-1}$ (Pitts et al., 2018), so that only CALIPSO nighttime measurements are used to analyze PSCs (Pitts et al., 2009). Höpfner et al. (2009) discussed
several effects, e.g. horizontal shifts of the PSC relative to the tangent point, the horizontal cloud extent, and diffuse boundaries as possible causes for the underestimation. Yet none of these reasons could explain the variations with altitude (Höpfner et al., 2009). They speculated that rather patchier PSC structures at higher altitudes are causing the higher underestimations than an altitude dependent sensitivity towards PSCs. However, based on our results a low PSC extinction coefficient alone can be sufficient to explain underestimations of the cloud top height of up to 2.5 km in the UTLS and even more at altitudes above
about 21 km, where the MIPAS vertical sampling is increased to 3 km.





For the comparison of tropospheric clouds between $50°$ N and $50°$ S and 12 to 20 km altitude Sembhi et al. (2012) calculated 3 monthly averages for $5°$ latitude and $10°$ longitude grid boxes using the altitude and latitude dependent CI thresholds that are also sensitive to PSCs and aerosol. However, to exclude volcanic aerosol Sembhi et al. (2012) selected two 3 monthly intervals relatively free of volcanic aerosol, JJA 2007 and DJF 2007/2008. This means that this data set mainly contains

optically thick clouds in mid-IR and a smaller fraction of optically thin clouds around the tropopause. From our simulations cloud top height differences between $-0.2$ and 1.6 km can be expected for optically thick clouds with extinction coefficients larger than $5 \times 10^{-2}$ km$^{-1}$. This is in agreement with the differences to CALIOP and HIRDLS found by Sembhi et al. (2012). The differences in the 3 monthly averaged grid boxes reached up to 1 km compared to CALIOP and on average the cloud top heights were 0.75 km higher for MIPAS than for CALIOP and HIRDLS.

In another comparison of tropospheric clouds between $50°$ N and $50°$ S Spang et al. (2012) investigated matches between MIPAS and GLAS lidar profiles between September and November 2003, when the MIPAS vertical sampling was 3 km. The results, given as mean differences in 3 km altitude bins, show that MIPAS systematically overestimated the cloud top height by mostly less than 2 km (Spang et al., 2012, Fig. 18). Based on our results, we expected an overestimation for tropospheric clouds of up to 1.6 km. In the same study also PSCs were included at latitudes poleward of $50°$. For PSCs overestimations of

the cloud top height of more than 5 km were found compared to GLAS. Such a high overestimation is in clear contradiction to the tendency towards underestimation found by Höpfner et al. (2009) for PSCs with respect to CALIOP and also cannot be explained by field-of-view effects. Spang et al. (2012) suggested a smaller sensitivity of GLAS measurements towards optically thin PSCs and inhomogeneities in the PSC fields as causes for the larger differences. Using less sensitive cloud detection methods for MIPAS the overestimation compared to GLAS was reduced (Spang et al., 2012). Hence, we consider

GLAS measurements not sufficiently sensitive toward optically thin clouds to be useful for a comparison with MIPAS cloud measurements.

For the particular case of sub-visible cirrus Spang et al. (2012) compared different cloud detection algorithms for MIPAS to SAGE II sub-visible cirrus detections. The comparison of zonal 3-monthly means measured in 2003 showed that outside the polar vortex most methods systematically underestimated the cloud top height by up to 2.5 km compared to SAGE II (Spang

et al., 2012, Fig. 12). Inside the polar vortex the MIPAS data include PSCs, whereas the SAGE II sub-visible cirrus data set does not. Only the MIPAS cloud top height based on altitude and latitude variable thresholds by Sembhi et al. (2012) overestimated the SAGE II cloud top heights in many cases. This overestimation, however, was attributed to an underestimation of the cloud occurrence at lower altitudes that lead to a higher average cloud top height for this method (Spang et al., 2012). Both, the over-, and the underestimations were considered to be within the range of the vertical sampling that was 3 km in 2003. Yet,

considering the extinction coefficients that are between $3 \times 10^{-4}$ and $3 \times 10^{-2}$ km$^{-1}$ for SAGE II sub-visible cirrus, we argue that a systematic underestimation of cloud top height by MIPAS can also be caused by the rather small extinction coefficients of sub-visible cirrus. In contrast to sulfate aerosol the SAGE II 1020 nm extinction coefficients for ice can be scaled to the mid-IR using a scaling factor of approximately 1. As Fig. 1 shows, top height underestimations of up to 2.4 km are possible for a vertical sampling of 1.5 km, which increases to about 3.9 km at 3 km vertical sampling while only overestimations of up

to 1.6 km are possible. Spang et al. (2012) compared the cloud top heights of individual profiles and presented the results in



3 km altitude bins. They found an overestimation of up to 1 km in the topmost altitude bin, and underestimations of up to 1, 3, and 5 km in the following altitude bins below. Based on our simulations all these values are well within the range that can be expected for the extinction coefficients of sub-visible cirrus.

## 6 Conclusions

In this study we characterized the aerosol and cloud top height information from Envisat MIPAS measurements. In the first step radiative transfer simulations that account for scattering on ice, volcanic ash, and sulfate aerosol particles were performed and evaluated. In the simulated scenarios MIPAS measurements were shown to be sensitive to sulfate aerosol down to an extinction coefficient of $1 \times 10^{-4} \mathrm{km}^{-1}$ at $10.5\,\mu$m. The sensitivity was positively correlated with the vertical thickness of the cloud layer. A larger vertical extent of the cloud leads to a smaller lower detection threshold due to a larger fraction covered with

cloud within MIPAS' vertical field-of-view. The dependency of the detection sensitivity on the extinction coefficient and cloud covered field-of-view fraction affected the derived cloud top height. For optically thick clouds ($\beta_e \geq 5 \times 10^{-3}\ \mathrm{km}^{-1}$) , such as cirrus clouds that lead to a constant ACI profile at all altitudes below the cloud, the cloud top height can be overestimated by up to 1.6 km due to MIPAS' broad field-of-view. In contrast, optically thin clouds ($\beta_e \leq 1 \times 10^{-3}\ \mathrm{km}^{-1}$), such as sulfate aerosol, PSCs, and sub-visible cirrus that cause a pronounced minimum in the ACI profile with minimum values of about 5.5

or larger, the cloud top height can be underestimated by up to 5.1 km, which is a combination of the field-of-view effect and the low extinction. For minimum ACI values between 3 and 5.5 over- and underestimation of the aerosol/cloud top height are possible. Further, MIPAS' coarse vertical sampling of 1.5 km contributes to the large derived top height uncertainty ranges.

In the second step MIPAS measurements of volcanic sulfate aerosol from the Nabro eruption were compared with lidar and twilight measurements. MIPAS detected the Nabro aerosol between the eruption in June 2011 until the end of its lifetime in

April 2012. Nabro aerosol measurements were available from June to August from CALIOP single profiles, in July and August 2011 from twilight measurements, and between July 2011 to February 2012 from ground based lidar measurements. The lidar and twilight data show that while the average Nabro aerosol extinction coefficient (@ 532 nm) in June and July was larger than $3 \times 10^{-3}\ \mathrm{km}^{-1}$ and the layer thickness was mostly below 4 km, the average extinction coefficient decreased with time and the layer thickness increased to 9.5 km (Fig 9)

Making the inherently different measurements of the aerosol layer top height comparable required an assessment of the sensitivities of the various top height estimation methods. Therefore Mie scattering simulations for measured volcanic sulfate aerosol particle size distributions were used to scale the MIPAS IR detection threshold ($\beta_{e\_thresh} = 1 \times 10^{-4}\ \mathrm{km}^{-1}$) to the lidar and twilight wavelengths. For sulfate aerosol the extinction coefficient in the visible is more than one order of magnitude (a factor of about 30) larger than in the mid-IR at $10.5\,\mu$m. Hence, an altitude variable extinction threshold (Winker et al., 2009)

that has an extinction coefficient of $3 \times 10^{-3}\ \mathrm{km}^{-1}$ at 18 km was used as a threshold for the cloud top height derived from CALIOP. This threshold was confirmed by a sensitivity test analyzing the number of matches between CALIOP and MIPAS aerosol detections for varying thresholds. For up to 2 months of Nabro aerosol measured by CALIOP, the commonly used gradient method (Mattis et al., 2008) for aerosol layer top estimation from lidar measurements resulted in 55.1 % in the same





($\pm 0.2$ km) top height as the extinction coefficient threshold of $3 \times 10^{-3}$ km$^{-1}$, while in 7.7 % the top height was higher and in

37.2 % it was lower. In contrast, for the aged volcanic aerosol plume measured by the ground based lidars the average plume extinction was below $3 \times 10^{-3}$ km$^{-1}$ and here the gradient method led to significantly higher top heights or even allowed for the detection of thinner layers that would have been missed using the extinction threshold.

As the extinction coefficient of the Nabro plume was low in general, MIPAS underestimated the aerosol layer top height. Compared to the CALIOP measurements that due to its lower sensitivity limit detected the plume mostly in the first two months

after the eruption, the MIPAS measurements underestimated the aerosol layer top height in $84-95$ % of all matches by 0.9 km in average. For the ground based lidar measurements of the aged and dispersed plume with extinction coefficients below $3 \times 10^{-3}$ km$^{-1}$ (@ 532 nm, which corresponds to $1 \times 10^{-4}$ km$^{-1}$ at $10.5 \mu$m), the MIPAS measurements always underestimated the aerosol layer top height by at least 0.9 up to 7.2 km. Compared to the twilight measurements that have an overlap with the CALIOP measurements the MIPAS measurements underestmated the cloud top height by up to 5.2 km in all but one case,

where an overestimation of 0.3 km was found. The results of this comparison are consistent with each other and in good agreement with the simulations predicting the underestimation of the cloud top height for low extinctions. Moreover, the comparison indicates that MIPAS can be sensitive to even lower extinctions ($< 1 \times 10^{-4}$ km$^{-1}$ at $10.5 \mu$m) if the vertical thickness of the cloud layer is larger than the 6 km considered in the simulations.

Our results show that in addition to the known causes for cloud top height uncertainties in IR limb emission measurements,

namely the overestimation due to a large field-of-view and the underestimation due to broken cloud conditions, also the extinction coefficient of the aerosol/cloud layer has an impact on the derived cloud top height. Previous studies showing MIPAS over- and/or underestimating cloud top heights were found to be not contradictorily, but rather complementary to each other as they investigated different cloud types covering a large range of characteristic extinction coefficients. The over- and underestimations of cloud top height in the previous studies can be explained by the effect of the cloud extinction coefficient and the

vertical field-of-view. Since for MIPAS measurements reliable algorithms to discriminate between ice clouds and aerosol and to filter out optically thick profiles are available the altitude uncertainty ranges can be narrowed down for each group. Although MIPAS' vertical sampling is relatively coarse and the altitude uncertainty is large compared to e.g. CALIPSO and SAGE II, MIPAS' vertically resolved aerosol and cloud measurements provide additional information by covering the entire Earth from pole to pole at day- and nighttime and even fill gaps in the sensitivity towards aerosol and cloud particles covering a wide range

of extinctions.

*Code and data availability.* JURASSIC is freely available under https://jugit.fz-juelich.de/slcs/jurassic-scatter. Refractive indices used for the simulations are available from the HITRAN compilation at http://hitran.org. The MIPAS level 1b IPF version 7.11 data can be accessed after registration via ESA's Earth Online portal https://earth.esa.int/web/guest. The MIPAS CI, ACI, aerosol, ash, and cloud detections shall soon be available at https://datapub.fz-juelich.de/slcs.



*Author contributions.* SG designed and performed the study. LH and RS contributed to the setup and analysis of the radiative transfer simulations for MIPAS. PS prepared and provided the Leipzig lidar data. CR prepared and provided the Jülich lidar data. PA prepared and provided the ESRANGE lidar data. NM prepared and provided the Tbilisi twilight data. JPV prepared and provided the CALIOP lidar data. SG wrote the paper with contributions from all coauthors.

*Competing interests.* The authors declare that no competing interests are present.

*Acknowledgements.* The authors gratefully acknowledge the computing time granted through JARA-HPC on the supercomputer JURECA at Forschungszentrum Jülich. The authors thank Anu Dudhia for providing the Reference Forward Model (RFM) that was used to generate the look-up tables for JURASSIC. The MIPAS level 1b IPF version 7.11 data were provided by the European Space Agency (ESA) and the analysis of the simulation results was supported by the ESA contract No 400011677/16/NL/LvH within the framework of the project "Particulate matter in the upper troposphere and stratosphere".



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





**Table 1.** Overview of relevant instrument characteristics for global aerosol measurements. The minimum detection sensitivity is given for the listed wavelength. The maximum sensitivity indicates the point when the measurements run into saturation, i.e. the top of thicker clouds is always detectable. In the comments column also the references are given.

| instrument | channel | sensitivity range | vertical sampling | coverage | profiles per day | comments & references |
|---|---|---|---|---|---|---|
| CALIOP | 532 nm | min: $3 \times 10^{-3} -$ $1.1 \times 10^{-2}$ km$^{-1}$ | 0.2 km | 82°N – 82°S | $\sim$ 3500 | **nighttime** backscatter ratio for 80 km averaged profiles at 15 km is about 1.5 km$^{-4}$sr$^{-1}$; lidar ratio range 20 – 70 sr$^{-1}$ (Winker et al., 2009); lidar ratio for sulfate rich volcanic aerosol is 60 – 70 sr$^{-1}$ (Prata et al., 2017) |
| | | min: $5 \times 10^{-3} -$ $1.8 \times 10^{-2}$ km$^{-1}$ | | 82°N – 82°S | $\sim$3500 | **daytime** backscatter ratio at 15 km is about $2.5 \times 10^{-3}$ km$^{-1}$sr$^{-1}$; lidar ratio range 20 – 70 sr$^{-1}$ (Winker et al., 2009) |
| | | min: $1 \times 10^{-3}$ km$^{-1}$ | | 50°N – 82°S 82°N – 50°S | $\sim$2800 | **nighttime** dedicated aerosol product; 1° averaged profiles (Vernier et al., 2009); lidar ratio 50 |
| GOMOS | 550 nm | min: $1 \times 10^{-4} -$ $1 \times 10^{-3}$ km$^{-1}$ max: $2 \times 10^{-2}$ km$^{-1}$ | 0.2 – 1.7 km | 87.5°N – 80°S | 110 | **nighttime** extinction (Vanhellemont et al., 2010; Sofieva et al., 2013; Robert et al., 2016; Vanhellemont et al., 2016); sensitivity depends on altitude (numbers derived from Fig. 2 in Vanhellemont et al. (2016)); the target of the aerosol retrieval is 4 km vertical resolution (Bertaux et al., 2010; Kyrölä et al., 2010); lowest tangent altitude 8 – 18 km high latitudes and tropics respectively (Tamminen et al., 2010) |
| IR nadir | 11 $\mu$m (909 cm$^{-1}$) | min: 0.01 km$^{-1}$ | – | 90°N – 90°S | – | **day- and nighttime** AOD needs to exceed 0.01 to make aerosol detectable with tri-spectral approach (Ackerman, 1997) |





| instrument | channel | sensitivity range | vertical sampling | coverage | profiles per day | comments & references |
|---|---|---|---|---|---|---|
| MIPAS | 12 $\mu$m (833 cm$^{-1}$) | min: $1 \times 10^{-5}$ km$^{-1}$ | 1.5 km | 89.3°N – 87.5°S daily | up to 1344 | **day- and nighttime** profiles (Fischer et al., 2008; Höpfner et al., 2009); lower detection limit only, derived from clear air simulations (including "background" aerosol) given in Sembhi et al. (2012) |
| | 10.5 $\mu$m (950 cm$^{-1}$) | min: $1 \times 10^{-4}$ km$^{-1}$ max: $1 \times 10^{-1}$ km$^{-1}$ | | | | simulated extinction coefficient range for sulfate aerosol covered $1 \times 10^{-4} - 1 \times 10^{-2}$ km$^{-1}$; ice cloud and volcanic ash get optically thick for $1 \times 10^{-1}$ km$^{-1}$ and higher (Griessbach et al., 2016) |
| OSIRIS | 750 nm | min: $4 \times 10^{-6}$ km$^{-1}$ – $4 \times 10^{-5}$ km$^{-1}$ max: $2 \times 10^{-3}$ km$^{-1}$ | 2 km | 82°N – 82°S 40°N/30°S during polar night | 100 – 400 | **daytime** extinction profiles (Rieger et al., 2015); lower limit given as absolute noise for 30 km ($4 \times 10^{-6}$ km$^{-1}$) and 10 km ($4 \times 10^{-5}$ km$^{-1}$) (Rieger et al., 2014), upper limit given in Sofieva et al. (2013) and Fromm et al. (2014) |
| SAGE-II | 1020 nm | min: $5 \times 10^{-6}$ km$^{-1}$ max: $2 \times 10^{-2}$ km$^{-1}$ | 1 km | 80°N – 80°S within a month | 30 | **twilight** extinction profiles (Wang et al., 1995; Thomason et al., 1997; Antuña et al., 2002; Thomason et al., 2008) with highest sensitivity down to 5 km; measurements at other wavelength 525, 386, 452 nm have lower sensitivity and reach down to 5, 16 and 12 km respectively (Thomason and Vernier, 2013) |



**Table 2.** Range of cloud top height uncertainty as a function of the extinction coefficient derived from the simulations. The cloud top is the altitude where ACI< 7. The results are given on the 0.1 km fine grid of the simulations and transferred to the 1.5 km vertical sampling of MIPAS. Please, see text for details.

| extinction coefficient in $km^{-1}$ | top height uncertainty on | |
|---|---|---|
| | fine grid in km | MIPAS grid in km |
| $1 \times 10^{-4}$ | $-3.7 - -1.0$ | $-5.1 - -1.0$ |
| $5 \times 10^{-4}$ | $-0.9 - 0.8$ | $-2.3 - 0.8$ |
| $1 \times 10^{-3}$ | $-1.0 - 1.1$ | $-2.4 - 1.1$ |
| $5 \times 10^{-3}$ | $0.7 - 1.4$ | $-0.3 - 1.4$ |
| $1 \times 10^{-2}$ | $1.0 - 1.5$ | $-0.5 - 1.5$ |
| $5 \times 10^{-2}$ | $1.2 - 1.6$ | $-0.2 - 1.6$ |
| $1 \times 10^{-1}$ | $1.0 - 1.5$ | $-0.5 - 1.5$ |
| $5 \times 10^{-1}$ | $1.2 - 1.6$ | $-0.2 - 1.6$ |
| $1$ | $1.3 - 1.6$ | $-0.1 - 1.6$ |





**Table 3.** Scaling factors for the extinction coefficient of sulfate aerosol from $948.5\,\mathrm{cm}^{-1}$ ($10.5\,\mu\mathrm{m}$, MIPAS) to the wavelengths used by selected satellite instruments, the ground-based lidar and twilight measurements, and for the MIPAS CI.

| wavelength | scaling factor measured PSDs | scaling factor fitted PSDs | instrument |
|---|---|---|---|
| 355 nm | 31.3 – 59.4 | 31.5 – 69.5 | Jülich lidar |
| 532 nm | 24.7 – 37.9 | 30.4 – 45.0 | CALIOP, Leipzig, ESRANGE lidar |
| 550 nm | 22.3 – 43.4 | 24.5 – 36.5 | GOMOS |
| 750 nm | 13.2 – 28.3 | 12.7 – 30.8 | OSIRIS |
| 780 nm | 14.9 – 32.0 | 15.6 – 31.2 | Tbilisi twilight measurements |
| 1020 nm | 5.3 – 18.1 | 5.5 – 21.7 | SAGE II |
| 12 $\mu$m | 0.56 – 0.59 | 0.56 | MIPAS $833\,\mathrm{cm}^{-1}$ CI window and IR nadir wavelengths |



**Table 4.** Difference between MIPAS and CALIOP cloud top height of the Nabro sulfate aerosol for the extinction coefficient threshold method and the gradient method for a possible set of scaling factors determining the altitude variable extinction detection threshold, where the threshold value is given at 18 km.

| | CALIOP | | MIPAS−CALIOP: extinction coefficient method | | | MIPAS−CALIOP: gradient method | | |
| scaling factor | threshold in $km^{-1}$ | # of matches | median $\Delta$ in km | mean $\Delta$ in km | underestimating fraction | median $\Delta$ in km | mean $\Delta$ in km | underestimating fraction |
|---|---|---|---|---|---|---|---|---|
| 20 | $2\times10^{-3}$ | 1143 | −1.4 | −1.5 | 97 % | −0.9 | −0.9 | 84 % |
| 30 | $3\times10^{-3}$ | 1121 | −1.1 | −1.2 | 95 % | −0.8 | −0.9 | 84 % |
| 40 | $4\times10^{-3}$ | 1083 | −0.9 | −0.9 | 91 % | −0.8 | −0.9 | 84 % |
| 50 | $5\times10^{-3}$ | 995 | −0.7 | −0.8 | 87 % | −0.8 | −0.8 | 85 % |





**Table 5.** Nabro sulfate aerosol measured by the Leipzig lidar and MIPAS. For the lidar data aerosol layer top, bottom, and mean extinction (for nighttimne profiles) are given. For MIPAS, the number of matching profiles, the mean cloud top height, and the corresponding minimum and maximum top heights are given.

| Profile | Date | Lidar top (km) | bottom (km) | extinction (km$^{-1}$) | # profiles | MIPAS top height mean (km) | min (km) | max (km) |
|---|---|---|---|---|---|---|---|---|
| 1 | 18.07.2011 | 11.0 | 9.1 | $4.2\times10^{-3}$ | 1 | 9.1 | 9.1 | 9.1 |
| 2 | 1.08.2011 | 16.3 | 13.6 | $2.1\times10^{-3}$ | 1 | 12.2 | 12.2 | 12.2 |
| 3 | 3.08.2011 | 16.3 | 13.6 | – | 2 | 14.6 | 13.8 | 15.4 |
| 4 | 3.08.2011 | 16.5 | 14.3 | – | 2 | 14.6 | 13.8 | 15.4 |
| 5 | 18.08.2011 | 18.0 | 15.5 | $3.6\times10^{-3}$ | 1 | 15.3 | 15.3 | 15.3 |
| 6 | 22.08.2011 | 18.0 | 14.7 | $3.7\times10^{-3}$ | 2 | 16.8 | 16.6 | 16.9 |
| 6 | | 10.0 | 7.1 | $2.6\times10^{-3}$ | | | | |
| 7 | 29.08.2011 | 16.5 | 11.9 | – | 3 | 13.8 | 12.2 | 15.2 |
| 8 | 30.08.2011 | 16.6 | 11.0 | – | 3 | 13.8 | 12.2 | 15.2 |
| 9 | 1.09.2011 | 16.2 | 12.1 | $3.4\times10^{-3}$ | 3 | 14.3 | 13.6 | 15.2 |
| 9 | | 9.6 | 8.5 | $1.9\times10^{-3}$ | | | | |
| 9 | | 7.1 | 3.9 | $2.0\times10^{-3}$ | | | | |
| 10 | 5.09.2011 | 17.5 | 12.1 | – | 5 | 14.7 | 13.5 | 15.4 |
| 11 | 12.09.2011 | 18.3 | 14.0 | $2.1\times10^{-3}$ | 1 | 15.1 | 15.1 | 15.1 |
| 12 | 12.09.2011 | 18.1 | 14.3 | – | 1 | 15.1 | 15.1 | 15.1 |
| 13 | 14.09.2011 | 17.6 | 12.9 | – | 1 | 15.0 | 15.0 | 15.0 |
| 14 | 15.09.2011 | 17.0 | 14.2 | $2.0\times10^{-3}$ | 1 | 13.8 | 13.8 | 13.8 |
| 14 | | 13.8 | 12.1 | $2.5\times10^{-3}$ | | | | |
| 15 | 15.09.2011 | 16.9 | 14.4 | – | 1 | 13.8 | 13.8 | 13.8 |
| 17 | 19.09.2011 | 18.3 | 11.5 | – | 4 | 14.4 | 13.5 | 15.2 |
| 18 | 26.09.2011 | 17.8 | 12.4 | – | 2 | 15.3 | 15.2 | 15.4 |
| 19 | 26.09.2011 | 18.1 | 12.8 | – | 2 | 15.3 | 15.2 | 15.4 |
| 20 | 29.09.2011 | 18.4 | 15.3 | $2.0\times10^{-3}$ | 1 | 15.3 | 15.3 | 15.3 |
| 20 | | 14.9 | 13.6 | $1.7\times10^{-3}$ | | | | |
| 20 | | 13.4 | 12.6 | $1.6\times10^{-3}$ | | | | |
| 21 | 30.09.2011 | 18.1 | 13.4 | – | 1 | 15.3 | 15.3 | 15.3 |
| 21 | | 13.4 | 12.6 | – | | | | |
| 22 | 1.10.2011 | 18.7 | 13.1 | $2.2\times10^{-3}$ | 1 | 15.3 | 15.3 | 15.3 |
| 23 | 23.10.2011 | 18.1 | 13.1 | – | 4 | 13.1 | 11.4 | 14.0 |
| 23 | | 13.0 | 11.7 | – | | | | |
| 23 | | 11.7 | 10.3 | – | | | | |
| 24 | 24.10.2011 | 18.2 | 15.0 | $1.9\times10^{-3}$ | 7 | 13.1 | 11.4 | 14.0 |
| 24 | | 14.7 | 10.1 | $3.0\times10^{-3}$ | | | | |
| 25 | 1.11.2011 | 17.3 | 14.0 | $2.0\times10^{-3}$ | 1 | 13.6 | 13.6 | 13.6 |
| 25 | | 13.9 | 9.9 | $2.0\times10^{-3}$ | | | | |
| 26 | 14.11.2011 | 17.8 | 12.3 | $2.9\times10^{-3}$ | 1 | 14.1 | 14.1 | 14.1 |
| 26 | | 12.0 | 10.3 | $2.9\times10^{-3}$ | | | | |
| 26 | | 10.0 | 8.3 | $2.6\times10^{-3}$ | | | | |
| 27 | 14.11.2011 | 17.6 | 11.3 | – | 1 | 14.1 | 14.1 | 14.1 |
| 27 | | 10.9 | 9.5 | – | | | | |
| 28 | 5.12.2011 | 16.2 | 12.3 | $2.5\times10^{-3}$ | 3 | 10.1 | 9.0 | 10.7 |
| 28 | | 11.8 | 8.2 | $4.1\times10^{-3}$ | | | | |
| 28 | | 7.1 | 6.0 | $3.6\times10^{-3}$ | | | | |
| 29 | 10.12.2011 | 17.2 | 13.8 | – | 1 | 10.5 | 10.5 | 10.5 |
| 29 | | 12.3 | 8.4 | – | | | | |
| 30 | 12.12.2011 | 16.3 | 12.4 | $2.4\times10^{-3}$ | 4 | 12.2 | 10.5 | 14.0 |
| 31 | 15.12.2011 | 16.0 | 8.0 | – | 4 | 10.5 | 9.4 | 10.9 |
| 33 | 2.02.2012 | 13.1 | 8.4 | – | 3 | 8.9 | 7.9 | 9.7 |
| 33 | | 8.3 | 7.2 | – | | | | |
| 33 | | 5.5 | 4.4 | – | | | | |
| 34 | 2.02.2012 | 11.8 | 8.5 | $3.2\times10^{-3}$ | 3 | 8.9 | 7.9 | 9.7 |





**Table 6.** Cloud top heights of the Nabro sulfate aerosol measured by the Leipzig lidar using the gradient method and the extinction threshold method.

| top height definition date | gradient method | extinction threshold $3 \times 10^{-3}\,\text{km}^{-1}$ |
| --- | --- | --- |
| 22.08.2011 | 18.0 km | 17.3 km |
| 24.10.2011 | 18.2 km | 15.1 km |
| 10.02.2011 | 12.9 km | 6.9 km |


**Table 7.** Nabro sulfate aerosol measured by the Jülich lidar and MIPAS. For the lidar data aerosol layer top, bottom, and mean extinction at 355 nm are given. For MIPAS, the number of matching profiles, the mean cloud top height, and the corresponding minimum and maximum top heights are given.

| Profile | Date | Lidar | | | MIPAS top height | | | |
| --- | --- | --- | --- | --- | --- | --- | --- | --- |
| | | top (km) | bottom (km) | extinction ($km^{-1}$) | # profiles | mean (km) | min (km) | max (km) |
| 1 | 01.08.2011 | 16.9 | 10.1 | $5.2\times10^{-3}$ | 2 | 14.7 | 13.8 | 15.6 |
| 2 | 17.08.2011 | 17.1 | 11.6 | $1.3\times10^{-2}$ | 4 | 14.9 | 13.6 | 15.5 |
| 3 | 17.08.2011 | 17.9 | 11.3 | $1.6\times10^{-2}$ | 1 | 15.3 | | |
| 4 | 18.08.2011 | 17.3 | 11.0 | $9.5\times10^{-3}$ | 4 | 13.9 | 12.4 | 15.3 |
| 5 | 23.08.2011 | 18.0 | 10.7 | $1.1\times10^{-2}$ | 2 | 16.9 | 16.8 | 17.1 |
| 6 | 24.08.2011 | 18.2 | 11.4 | $5.9\times10^{-3}$ | 2 | 16.9 | 16.8 | 17.1 |
| 7 | 12.12.2011 | 16.3 | 9.5 | $9.4\times10^{-3}$ | 2 | 11.5 | 10.9 | 12.1 |





**Table 8.** Nabro sulfate aerosol measured by the Esrange lidar and MIPAS. For the lidar data aerosol layer top, and bottom are given. For MIPAS, the number of matching profiles, the mean cloud top height, and the corresponding minimum and maximum top heights are given.

| Profile | Date | Lidar | | | MIPAS top height | | | |
|---|---|---|---|---|---|---|---|---|
| | | top (km) | bottom (km) | extinction (km$^{-1}$) | # profiles | mean (km) | min (km) | max (km) |
| 1 | 13.01.2012 | 14.06 | 10.16 | $6.1 \times 10^{-3}$ | 4 | 11.6 | 11.2 | 12.6 |
| 2 | 19.01.2012 | 15.56 | 8.06 | – | 1 | 12.8 | | |
| 3 | 23.01.2012 | 11.36 | 8.21 | $1.05 \times 10^{-2}$ | 1 | 9.8 | | |





**Table 9.** Nabro sulfate aerosol measured by the twilight technique, MIPAS, and CALIOP over Tbilisi. For the twilight data the aerosol layer top and bottom altitudes, the corresponding errors and the averaged extinction coefficient are given. For CALIOP the aerosol layer top, bottom, and average extinction are given. For MIPAS, the number of matching profiles, the mean cloud top height, and the corresponding minimum and maximum top heights are given.

| Profile | Date | Twilight top (km) | error (km) | error (km) | bottom (km) | error (km) | error (km) | extinction (km⁻¹) | Lidar top (km) | bottom (km) | extinction (km⁻¹) | MIPAS top height # profiles | mean (km) | min (km) | max (km) |
|---|---|---|---|---|---|---|---|---|---|---|---|---|---|---|---|
| 1 | 14.07.2011 | 22.6 | 1.1 | -1.2 | 16.6 | 1.1 | -0.9 | $4.1\times10^{-3}$ | 19.2 | 15.0 | $5.9\times10^{-3}$ | 2 | 18.3 | 17.4 | 19.2 |
| 2 | 18.07.2011 | 20.9 | – | – | 16.7 | – | – | $1.4\times10^{-3}$ | 18.8 | 14.8 | $4.3\times10^{-3}$ | 4 | 17.8 | 17.5 | 18.0 |
| 3 | 22.07.2011 | 17.5 | – | – | 11.9 | – | – | 8.1e-05 | 18.6 | 15.8 | $2.7\times10^{-3}$ | 4 | 17.1 | 16.0 | 17.8 |
| 4 | 27.07.2011 | 19.5 | 1.9 | – | 15.2 | – | – | $2.6\times10^{-3}$ | – | – | – | 1 | 17.3 | 17.3 | 17.3 |
| 5 | 28.07.2011 | 19.1 | 0.9 | -1.0 | 15.7 | 1.0 | -0.6 | $3.1\times10^{-3}$ | – | – | – | 1 | 17.3 | 17.3 | 17.3 |
| 6 | 29.07.2011 | 18.7 | 0.9 | -1.4 | 15.8 | 1.1 | -0.4 | $3.0\times10^{-3}$ | 19.0 | 16.2 | $1.4\times10^{-3}$ | 4 | 17.5 | 17.2 | 17.8 |
| 7 | 30.07.2011 | 19.3 | 1.2 | -1.2 | 15.7 | 1.1 | -1.0 | $2.9\times10^{-3}$ | 19.0 | 16.2 | $1.4\times10^{-3}$ | 4 | 17.5 | 17.2 | 17.8 |
| 8 | 01.08.2011 | 19.4 | – | – | 16.2 | – | – | 9.8e-04 | 18.6 | 16.4 | $1.9\times10^{-3}$ | 3 | 17.6 | 17.2 | 18.2 |
| 9 | 02.08.2011 | 19.2 | 2.0 | – | 15.7 | – | – | $1.8\times10^{-3}$ | 18.2 | 16.0 | $1.5\times10^{-3}$ | 3 | 17.6 | 17.2 | 18.2 |
| 10 | 03.08.2011 | 18.7 | – | – | 15.6 | – | – | $1.4\times10^{-3}$ | – | – | – | 1 | 17.6 | 17.6 | 17.6 |


**Figure 1.** Characterization of the cloud top height derived from simulations. a) Volume of the field-of-view filled with cloud at the detected cloud top height as a function of extinction. The colours indicate the particle type: orange - sulfate aerosol, blue - ice, and black - volcanic ash. b) top height difference (detected − real top height) as a function of extinction. c) Minimum ACI value within a cloudy profile as a function of extinction coefficient. d) top height difference as a function of minimum ACI value within a cloudy profile. In all panels the symbols indicate the particle type: circles - sulfate aerosol, diamonds - ice, squares - volcanic ash. In panels b), c), and d) the colours indicate the background atmosphere type: red - tropics, orange - midlatitudes, light blue - polar summer, and dark blue - polar winter. In panels b) and d) the colored symbols show the top height uncertainties of all scenarios for the simulated 100 m vertical sampling grid. For the MIPAS vertical sampling of 1.5 km the colored symbols indicate the upper limit and the grey symbols indicate the lower limit of the top height uncertainty range. Depending on the position of the cloud relative to the tangent altitude the detected top height may fall anywhere inbetween. See text and Table 2 for details.





**Figure 2.** Extinction coefficient spectra normalized to 1 at 948.5 cm$^{-1}$ for modelled sulfate aerosol particle size distributions (colored lines) and measured size distributions after Pinatubo (black dashed lines Deshler et al., 1992a, b, 1993, Arctic) and Nabro (grey dashed line Bourassa et al., 2012b, July, Wyoming). The corresponding scaling factors are given in Table 3.

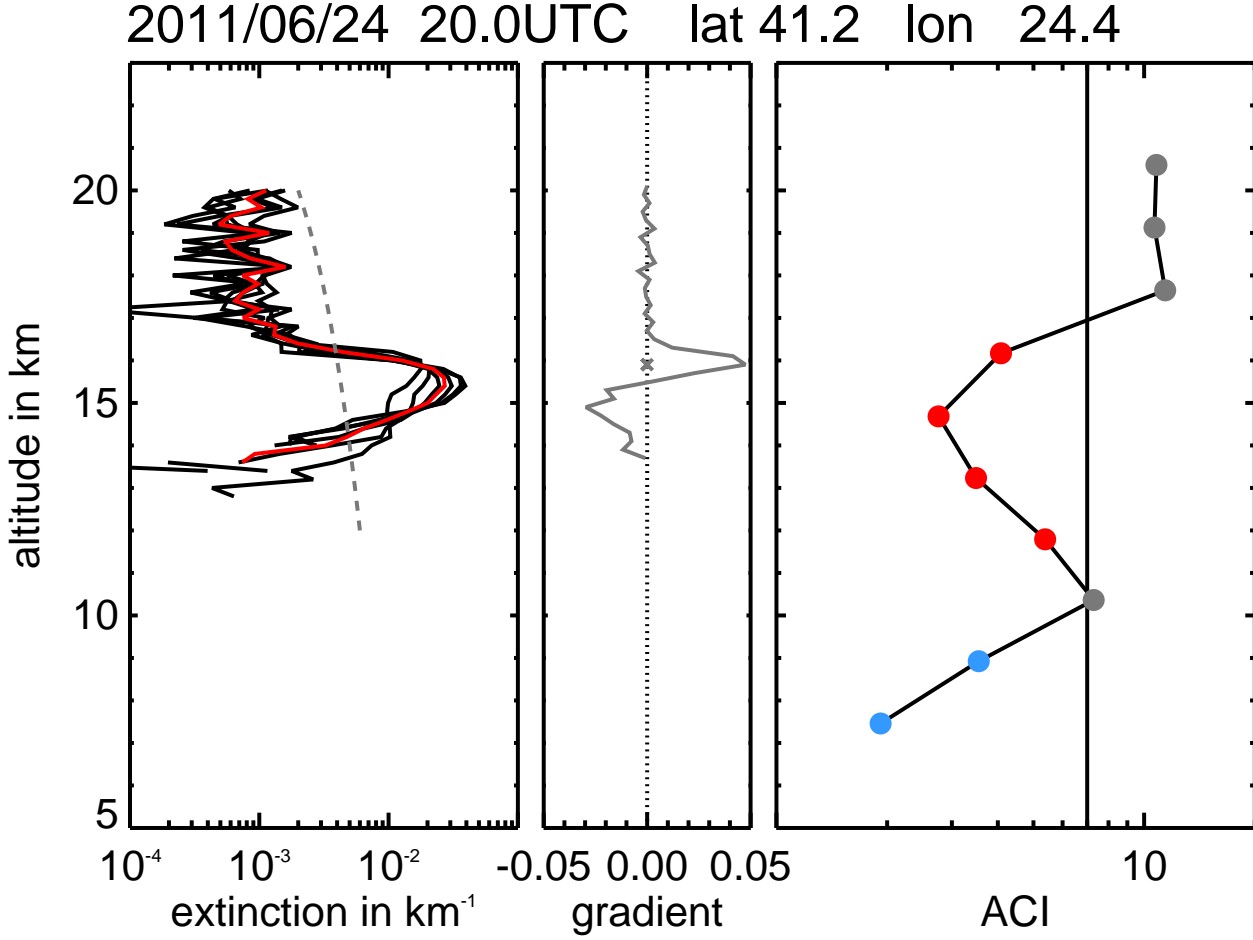

**Figure 3.** Nabro sulfate aerosol layer measured by CALIOP and MIPAS. Left: The red line is the averaged lidar profile of the individual lidar profiles within the match range (black lines). The grey dashed line indicates the altitude variable extinction coefficient threshold of $3 \times 10^{-3}$ km$^{-1}$ at 18 km used to estimate the top height of the aerosol layer (Winker et al., 2009). Middle: The gradient of the averaged lidar profile, where the cross at the maximum gradient indicates the top height according to the gradient method described by Mattis et al. (2008). Right: MIPAS ACI profile. The dots mark the tangent altitudes, where grey dots indicate clear air, red dots aerosol, and blue dots ice clouds. The black solid line marks the ACI detection threshold of 7.





**Figure 4.** Number of matches for MIPAS and CALIOP aerosol detections as a function of the altitude variable extinction threshold for CALIOP given at 18 km.



**Figure 5.** Sulfate aerosol detected by MIPAS (circles) and CALIOP (squares) a) 10 days and b) 48 days after the Nabro eruption. The MIPAS orbit tracks, indicated by grey dashes reach up to 90° N and the CALIOP orbit tracks of available aerosol data are inidcated by black dots. The colours indicate the layer top height. For CALIOP aerosol detections an altitude variable extinction threshold that is $3 \times 10^{-3}$ km$^{-1}$ at 18 km was used.







**Figure 6.** Distribution of cloud top height differences (MIPAS−CALIOP) for aerosol measurements of the Nabro sulfate aerosol in June, July, and August 2011. The cloud top height from CALIOP was derived using an altitude variable extinction threshold that is $3 \times 10^{-3}$ km$^{-1}$ at 18 km. The black dashed line indicates the median, the dotted lines are the 25 and 75 percentiles, and the solid line marks the zero difference.





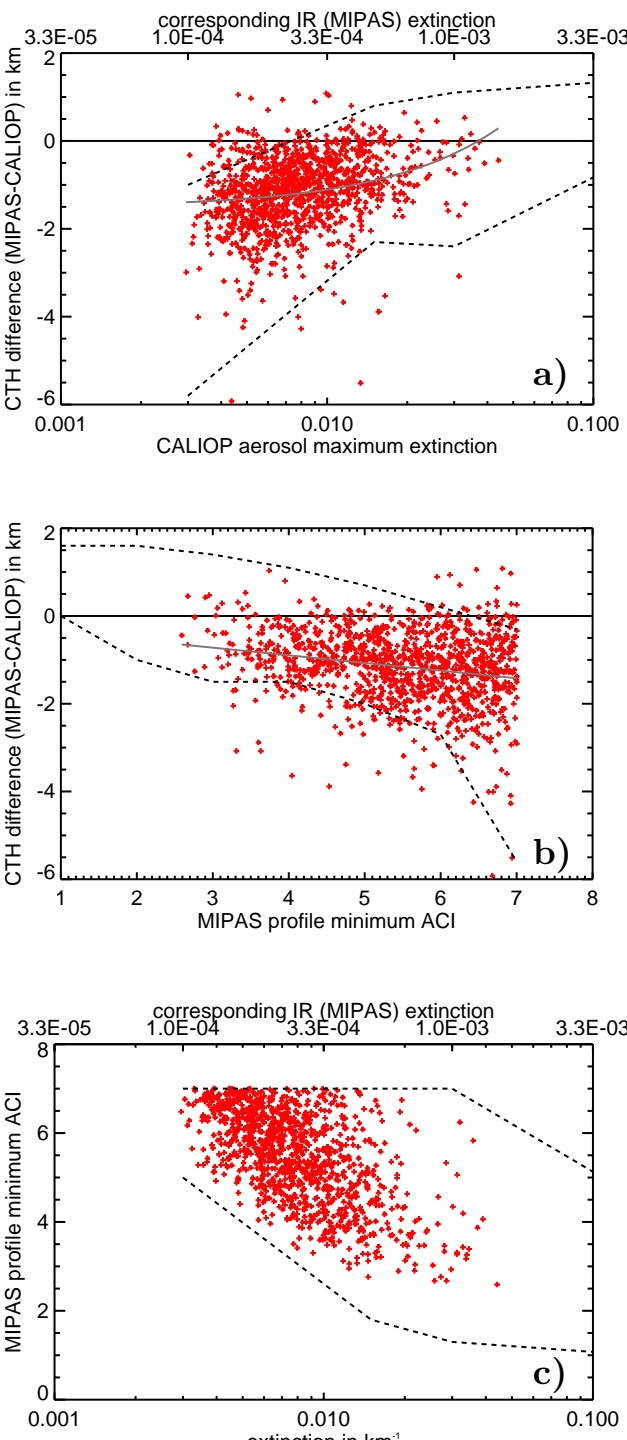

**Figure 7.** Cloud top height differences between MIPAS and CALIOP Nabro aerosol measurements as a function of a) CALIOP profile maximum extinction, and b) MIPAS minimum ACI. c) shows the diagnostic relation between extinction and MIPAS minimum ACI. The black dashed lines indicate the ranges expected from the simulations (see Fig. 1).

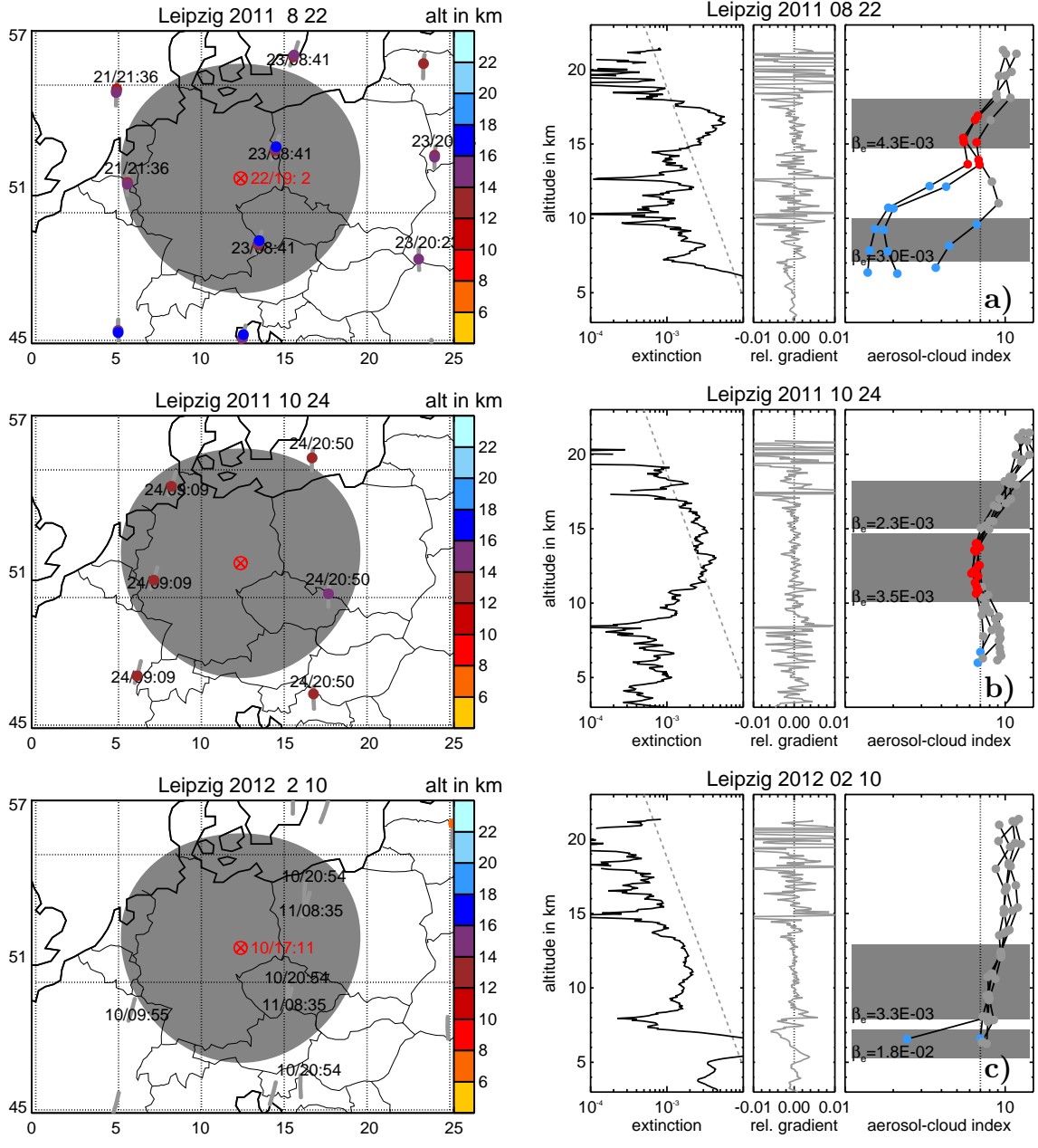

**Figure 8.** Nabro aerosol measured by the Leipzig lidar and MIPAS on a) 22 August 2011, b) 24 October 2011, and c) 10 February 2012. Left: The maps show the lidar station, lidar measurement time, match radius, MIPAS orbits, MIPAS profile measurement time, and MIPAS cloud top height (color coded). Middle: Leipzig lidar extinction profile (black line) and the relative extinction gradient (grey line). The extinction coefficient threshold of $3 \times 10^{-3}$ km$^{-1}$ at 18 km is indicated by the grey dashed line. Right: MIPAS ACI profiles within the match range. The grey dots denote clear air, red dots denote sulfate aerosol, and blue dots denote ice and optically thick clouds. The black dotted line is the ACI threshold of 7.



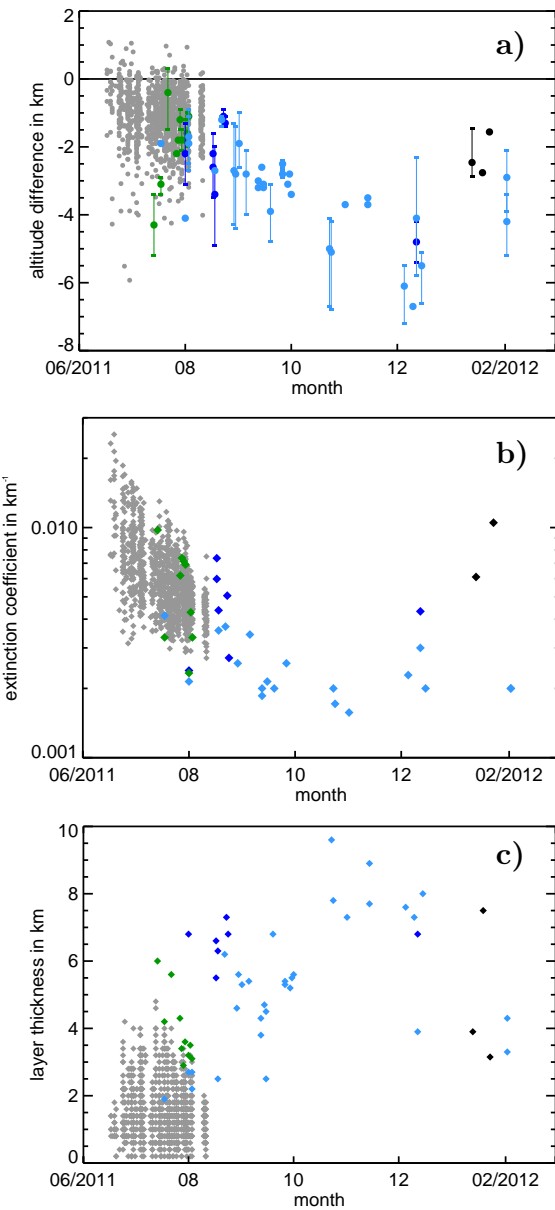

**Figure 9.** Properties of the Nabro sulfate aerosol as a function of time derived from CALIOP (grey), Leipzig lidar (light blue), Jülich lidar (dark blue), Esrange lidar (black), and twilight measurements (green). The CALIOP measurements cover the latitude range between 0 and 50°N and all ground based lidar stations are north of 50°N. Only for the twilight measurements there is a spatial and temporal overlap with CALIOP. a) Difference between the cloud top heights measured by MIPAS and CALIOP, the ground-based lidars, and the twilight measurements (MIPAS−lidar/twilight) during the 8 months after the Nabro eruption. If more than one MIPAS profile was within the match range, the circles indicate the average and the bars the range of the individual profiles. b) Nabro aerosol layer extinction coefficient at 532 nm derived from the lidar and twilight measurements between June 2011 to February 2012. For the Jülich lidar and the twilight measurements the data was scaled to 532 nm. c) Nabro aerosol layer thickness derived from the lidar and twilight measurements from June 2011 to February 2012.





**Figure 10.** Nabro aerosol measured by the Jülich lidar and MIPAS on a) 24 August 2011 and b) 12 December 2011. Left: The maps show the lidar station, lidar measurement time, match radius, MIPAS orbits, MIPAS profile measurement time, and MIPAS cloud top heights (color coded). Right: Jülich lidar extinction profile at 355 nm (black line) between aerosol top and bottom altitude (grey box) and MIPAS ACI profiles. In the MIPAS ACI profiles the grey dots denote clear air, red dots sulfate aerosol, and blue dots denote ice and optically thick clouds. The black dotted line is the ACI threshold of 7.





**Figure 11.** Nabro aerosol measurements over Esrange on a) 13.01.2012 and b) 20.01.2012. Left: Map with lidar station, lidar measurement time, match radius, MIPAS orbits, MIPAS profile measurement time, and MIPAS cloud top heights (color coded). Right: MIPAS ACI profiles and lidar backscatter ratio profiles. The grey areas denote the aerosol layer measured by the lidar, the black solid line is the parallel backscatter ratio and the black dashed line is the perpendicular backscatter ratio. In the MIPAS ACI profiles the grey dots denote clear air, red dots sulfate aerosol, and blue dots denote ice and optically thick clouds. The black dotted line is the ACI threshold of 7.





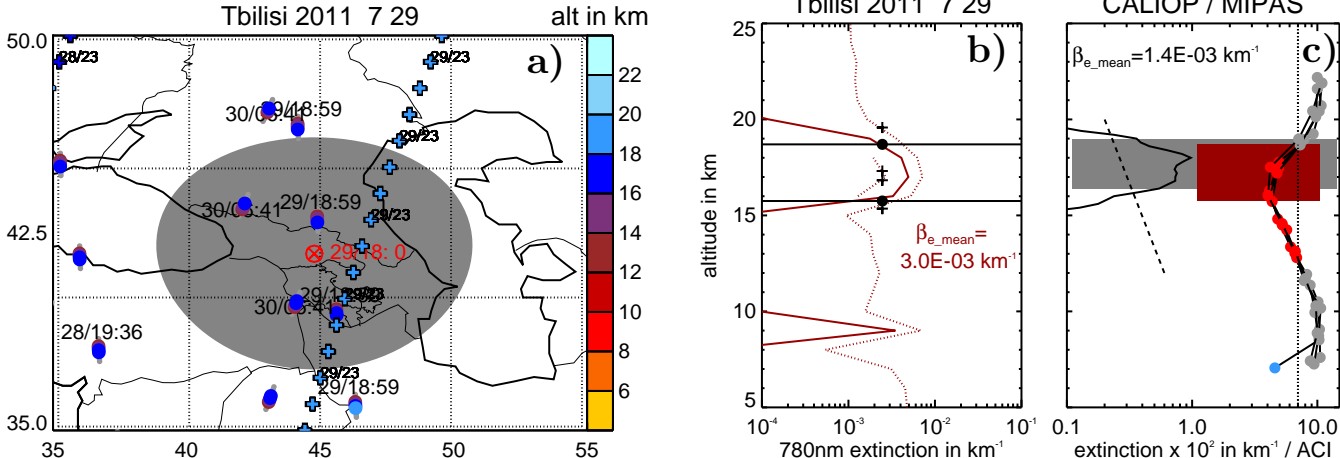

**Figure 12.** Measurements of Nabro aerosol over Tbilisi on 29 July 2011. a) Map with the twilight station, match radius, MIPAS and CALIPSO orbits, their measurement times, and the MIPAS (circles) and CALIOP (crosses) cloud top heights (color coded). b) Twilight extinction profile at 780 nm (red solid line) and uncertainty (red dotted lines). The black circles indicate the aerosol layer top and bottom altitude and the black crosses indicate the altitude uncertainty range. c) CALIOP extinction (black solid line) and MIPAS ACI profiles (black lines with colored circles). For the CALIOP profile the black dashed line indicates the extinction threshold and the grey box is the aerosol layer derived from CALIOP. For the MIPAS profiles grey dots denote clear air, red dots sulfate aerosol, and blue dots denote ice and optically thick clouds. The black dotted line is the ACI threshold 7. The red box denotes the aerosol layer detected by the twilight measurement.



## detection sensitivity to sulfate aerosol

**Figure 13.** Detection sensitivity to sulfate aerosol extinctions for different satellite based instruments. The blue bars indicate a conservative estimate of detectable extinctions given in Table 1 and scaled to $950\,cm^{-1}$. The gray bars indicate detectable but optically thick extinctions. The blue stripes indicate the uncertainties due to the scaling factor for SAGE II and IR nadir. For MIPAS, CALIOP, GOMOS, and OSIRIS the blue stripes are a combination of uncertainties due to the scaling factor and less conservative detection thresholds (lower minimum extinctions) given in literature. For CALIOP only the nighttime extinctions are considered. Further details on the instruments' measurement capabilities and the corresponding references are given in Table 1.