# Peer review of "Aerosol and cloud top height information of Envisat MIPAS measurements"

_Atmospheric Measurement Techniques, 2019_

## Referee Comment (RC1) · Anonymous Referee #1 · 10 Sep 2019

Review of: Aerosol and cloud top heigt information of Envisat MIPAS measurements

by Griessbach et al. August 2019

Overview

The estimate of cloud (or aerosol) height from MIPAS spectra follows a fairly crude system of identifying the highest altitude spectrum which has cloud-like continuum features and assigning that nominal altitude as the cloud top height.

The paper starts with model simulations for a variety of aerosols which demonstrate that this method will overestimate the height for optically thick aerosols and underestimate the height for optically thin aerosols (as would be expected), but also that this does not depend significantly on the type of aerosol. It is argued that this can explain

the seemingly contradictory results found in previous studies of MIPAS cloud/aerosol heights.

The main part of the paper then focuses on measurements of sulfate aerosol from the Nabro eruption in 2011, comparing the MIPAS-retrieved aerosol height with colocated measurements from the CALIOP instrument as well as various ground-based observations. Considerable attention has been paid to defining the top of the aerosol cloud in each of the datasets, as well as the scaling of measured extinction to infrared region observed by MIPAS.

The general conclusion is that, as the plume is optically thin, MIPAS underestimates the height and as the plume ages, becoming even thinner, the discrepancy gets worse. A useful by-product of this study is an assessment of the sensitivity of the various instruments to the detection of sulfate aerosol.

General Comments

1) Assuming any detected cloud is assigned the nominal tangent height at the centre of the MIPAS FOV, I think it is fairly clear why thick cloud at the bottom of the FOV will result in an overestimate of about half the FOV width, ie 1.5km, which is confirmed by the points at the top right of Fig 1b. It is also clear why progressively thinner cloud will result in a gradually decreasing estimate of cloud top height until the cloud is so thin that no cloud is detected.

The chosen diagnostic, cloud volume in FOV, is difficult to interpret. If I understand its definition, then halving the horizontal width of the MIPAS FOV would halve the volume but still produce the same plot, suggesting an arbitrary scaling.

Noting that the detection sensitivity depends on both the extinction coefficient and cloud-covered field-of-view fraction (P20 L625), a better diagnostic might be something like average aerosol absorption integrated over the MIPAS FOV, where absorption is defined as 1 - exp(-X) where X is the cloud extinction integrated along the line-of-sight

of any FOV sub-element. This would have a value between 0 for cloud free and 1 for thick cloud filling the FOV.

It also seems to be more naturally related to the threshold that actually triggers cloud detection. This would provide a more meaningful y-axis for Fig 1a and, I would expect, a more linear plot if used as the x-axis for Fig 1b. This would probably also explain the equivalence of the 1km v 6km layers discussed on P17 L530.

2) Given that the Nabro aerosol seems to lack a well-defined upper altitude, any comparison of height between two instruments will clearly be a sensitive function of the chosen thresholds. However while different thresholds are tried for the lidar instruments, the MIPAS value is fixed at ACI=7.

It would be interesting to see if higher ACI values yielded more information. For example, a plot of CALIOP values v ACI not only at the levels at which ACI reaches a value <=7 but also for the MIPAS tangent heights one or two levels above to see if there is any correlation.

3) Is the MIPAS instrument noise included in the sensitivity study? Also what is the impact of any residual radiometric offset after the calibration? Presumably this would have some effect on detection sensitivity if it produced a non-zero offset. Perhaps the Kleinert report has something on this?

Minor Comments

P1 L15: 'In the fresh to two months ...'. I would suggest rephrasing as 'For plumes up to two months old ...'

P2 L50: I am surprised that there is no mention of ACE/SCISAT among the solar occulation instruments. However, perhaps they just focus on deriving molecular concentrations.

P4 L96: 'index cloud detection' - should this be 'cloud index detection'?

P4 L119: Mention Envisat's sun-synchronous orbit so that the concent of a fixed mean local solar time makes sense.

P5 L125-135: Presumably the pairs of number in brackets refer to the wavenumber range rather than a pair of spectral points, so I'd suggest using '–' (LaTeX) rather than a comma. It would be helpful to have a diagram showing the spectra signatures being detected using these windows. Also, these all lie within the band A, so how is band B used?

P5 L136: 'that has flown ... since' rather than 'that flies ... since'.

P5 L149: Either 'between ... and ...' or 'from ... to ...'

P6 L166: 'on average' rather than 'in average'

P7 L189: Suggest 'where, potentially, cirrus clouds ...'

P7 L205: If using 'the' for dates, then 'the 4th and the 25th'

P8 L224: Either 'These ... are ...' or 'This ... is ...'

P9 L268: Comparing Fig 1b and Table 2 it looks as if Fig 1b shows the differences after transferring to the coarser MIPAS sampling, whereas the text seems to imply Fig 1b is before the transfer.

P14 L445: 'discrepancy'

P18 L562: 'contradictory'

Table 2: hard to distinguish between '-' indicating a range of values and a negative sign. In LaTeX use either '–' or '$-$'.

Fig 7: Not clear what CALIOP aerosol maximum extinction means. If this is the maximum value of extinction in the CALIOP profile (equivalent to that found by the subsequent gradient method) then that may be lower in altitude than the part of the profile which MIPAS is detecting. So perhaps something else is meant by 'maximum'?

[Figure]

Fig 8 caption: on the right hand plots, what are the grey boxes and beta numbers?

Fig 9 caption: 'Only for the twilight measurements is there ...'

Fig 13: given the standard representation is to scale aerosol extinctions to 550 or 532nm, I think it would be useful to add a second x-axis along the top with this scaling as well.

―――

―――――――――――――――――――――

---

## Referee Comment (RC2) · Anonymous Referee #2 · 2 Oct 2019

In this manuscript, Griessbach and co-workers present a study evaluating a commonly used method to determine cloud/aerosol top heights from MIPAS/Envisat thermal limb emission measurements. The study is based on simulations as well as on comparisons with a variety of independent measurements of the volcanic plume from the Nabro eruption by other satellite and ground-based instruments. The authors conclude that, in addition to the effects of an extended vertical field-of-view and of inhomogeneous cloud cover on the determination of cloud-top altitudes, the cloud/aerosol optical depth is a further important reason for uncertainty. It is suggested that the interplay of these three main contributors to the MIPAS cloud top height uncertainty resolves the puzzle of contradictory results from a variety of previous studies. As an add-on, a comparison of the sensitivity of different remote sensing techniques towards sulfate aerosol is
presented. This work fits well into the scope of AMT and I strongly support publication after a few specific comments are taken into consideration.

Specific comments:

MIPAS-simulations: To be able to estimate how much the instrumental performance itself contributes to the cloud/aerosol top height estimation, it would be interesting to estimate/discuss the errors introduced by the random and systematic uncertainties of MIPAS, e.g. spectral noise, radiometric accuracy, and others as described by Kleinert et al., 2018.

Could you discuss whether it would make sense with respect to detection sensitivity, to use the radiances at the maximum of the sulfate peak around 1100 cm-1 instead of those around 800 cm-1 for the aerosol detection since the absorption seems to be an order of magnitude higher?

Regarding the comparison with CALIOP: could the variability of the CALIOP aerosol top height within the match-criteria be used to estimate the plume's homogeneity at its upper level and be correlated with the MIPAS cloud-top in order to distinguish between cloud-inhomogeneity and optical thickness as the reason for the underestimation by MIPAS?

Throughout the paper it is argued with extinction. However, would a quantity like optical depth covered by the field-of-view not be better suited?

L534-542: It should be made clear that these considerations are valid for the typical size distribution of sulfate aerosols. Could you also consider/discuss cases for other particle sizes (e.g. smaller particles) where scattering in the UV/VIS is decreased but the absorption signal in the mid-IR is not/less affected?

Table 1: SCIAMACHY and OMPS NPP may be added. The first one since it could be directly compared to MIPAS in future work and the second to cover the present time and the future.

Technical comments:

L50: 'occulation' -> 'occultation'

L69: why is 'However' used here?

L125: 'color' vs. Figure 1 caption: 'colours', please harmonize

L269: 'maximal' -> 'maximum'

L306: 'compareable' -> 'comparable'

L416: delete ')'

L417: 'analysed' but also 'analyzed' is used

L445: 'dicrepancy' -> 'discrepancy'

L450: 'exinction' -> 'extinction'

L562: 'contradicory' -> 'contradictory'

L659: 'underestmated' -> 'underestimated'

L679: 'soon be available': is the dataset already available?

Table1, last column: '1.5 kmˆ-4 srˆ-1' -> '1.5ˆ-4 kmˆ-1 srˆ-1'

Fig. 5, caption: 'inidcated' -> 'indicated'

Fig. 9: could you also show a further panel with the absolute plume altitudes to better judge the difference compared to the absolute value.

Fig. 9: a legend, e.g. in one of the panels indicating the different instruments would be better than only having the information in the caption.

Fig. 13, caption: 'gray' -> 'grey'

[Figure]

---

## Author Comment (AC1) · 6 Dec 2019

**Reply to Reviewer 1**

We thank the reviewer for reviewing our manuscript and providing comments, questions, food for thoughts, and corrections. We went though all points and provide answers to each of them. Where necessary, we also modified the manuscript. Please see below for the details.

**General Comments:**

*1) Assuming any detected cloud is assigned the nominal tangent height at the centre of the MIPAS FOV, I think it is fairly clear why thick cloud at the bottom of the FOV will result in an overestimate of about half the FOV width, ie 1.5km, which is confirmed by the points at the top right of Fig 1b. It is also clear why progressively thinner cloud will result in a gradually decreasing estimate of cloud top height until the cloud is so thin that no cloud is detected.*

Yes, we agree, this is what we showed and quantified in our study. We showed, which extinctions can be considered optically thick for IR limb emission measurements leading to overestimations of cloud top height, which extinctions can be considered optically thin leading to systematic underestimations, and which cloud extinctions are too thin to be detectable. We also quantified the top altitude uncertainty ranges as a function of cloud extinction.

*The chosen diagnostic, cloud volume in FOV, is difficult to interpret. If I understand its definition, then halving the horizontal width of the MIPAS FOV would halve the volume but still produce the same plot, suggesting an arbitrary scaling.*

Maybe there is a misunderstanding here that we have to clarify. Actually it is not FOV-volume at tangent point, but FOV-volume in the cloud integrated along the LOS. In Section 3.2 (P9 L267 in the revised manuscript) we rephrased to:
"... For each detected aerosol/cloud top height Fig. 2a shows the cloud-filled field-of-view volume integrated along the line-of-sight as a function of aerosol/cloud extinction coefficient. ..."
In the following we changed "field-of-view volume" to "integrated cloudy field-of-view volume".

Concerning the horizontal extent of the cloud, we cannot draw any conclusions from our setup. In the simulations we made the assumption of an infinite homogeneous cloud layer that fills the entire horizontal MIPAS FOV. So yes, when changing the horizontal FOV, the volume would change too. But, to our knowledge the MIPAS horizontal FOV is assumed to have a fixed value of 30 km. If the cloud layer was not infinite (larger than 30 km) in the horizontal and would only fill part of the horizontal FOV, the integrated-cloudy-FOV-volume would become smaller. However, in our simulation setup we did not consider horizontal cloud inhomogeneities on the scale of 30 km. The only way to change the integrated cloudy FOV volume here, is through changing the vertical extent.

*Noting that the detection sensitivity depends on both the extinction coefficient and cloud-covered field-of-view fraction (P20 L625), a better diagnostic might be something like average aerosol absorption integrated over the MIPAS FOV, where absorption is defined as 1 - exp(-X) where X is the cloud extinction integrated along the line-of-sight of any FOV sub-element. This would have a value between 0 for cloud free*

*and 1 for thick cloud filling the FOV. It also seems to be more naturally related to the threshold that actually triggers cloud detection. This would provide a more meaningful y-axis for Fig 1a and, I would expect, a more linear plot if used as the x-axis for Fig 1b. This would probably also explain the equivalence of the 1km v 6km layers discussed on P17 L530.*

We have actually investigated this issue in a similar way as suggested. We started with a 2D setup and integrated the vertically oriented area filled with cloud for each tangent point (Fig 1a). In the following we used only the tangent heights of the detected cloud top heights. To extend the area to a (3D) volume, we assumed the 3D FOV as a rectangular tube and multiplied the area by 30 km. Fig 2a in the revised manuscript and Fig. 1b in the reply show the integrated FOV-volume along the LOS at the detected cloud top as a function of cloud extinction. Analogously to the 1D-parameter AOD (path length in cloud $\times$ extinction) we then calculated the 3D aerosol optical volume (AOV) (integrated-cloudy-FOV-volume-along-the LOS $\times$ extinction[3]). In contrast to the expectation that the AOV at cloud detection should be close to a constant value, the AOV at the detected cloud top shows an exponential dependency on extinction (Fig. 1c). Using AOV instead of extinction in Fig. 1d gives a very similar picture to Fig. 2b in the revised manuscript. Following your suggestion, we calculated the aerosol absorption as $1 - \exp(-\mathrm{AOV})$ (Fig. 1e). Fig. 1f shows that the altitude difference as a function of absorption ($1-\exp(-\mathrm{AOV})$) is very similar to the results presented as a function of extinction. Since the extinction is the dominant factor and allows for a better comparison with the full suite of other instruments, we prefer presenting the results as a function of extinction.

*2) Given that the Nabro aerosol seems to lack a well-defined upper altitude, any comparison of height between two instruments will clearly be a sensitive function of the chosen thresholds. However while different thresholds are tried for the lidar instruments, the MIPAS value is fixed at ACI=7. It would be interesting to see if higher ACI values yielded more information. For example, a plot of CALIOP values v ACI not only at the levels at which ACI reaches a value <=7 but also for the MIPAS tangent heights one or two levels above to see if there is any correlation.*

Actually the Nabro aerosol has a well-defined upper altitude as can be seen in the lidar measurements. It is the instruments measuring this aerosol layer that introduce some uncertainty. For MIPAS the uncertainty is introduced by the large FOV and coarse vertical sampling. For CALIOP the uncertainty is introduced by averaging the profiles over 1° latitude along track and sampling the data on a 0.2 km grid in the vertical. This averaging, however, has to be done to achieve a better signal-to-noise ratio, so that the CALIOP sensitivity becomes comparable to the MIPAS detection sensitivity using an ACI threshold of 7. So, for CALIOP, we tried different thresholds to avoid misinterpreting noise as an aerosol signal.
For MIPAS an ACI of 7 was chosen, because it has a comparable detection sensitivity to the most sensitive altitude variable CI thresholds derived by Sembhi et al. (2012) (Griessbach et al., 2016). This study does not aim at improving the detection sensitivity of aerosol and cloud detection methods, but rather on assessing the altitude information that existing methods provide.

Now, to the interesting part: MIPAS ACI profiles (Fig. 2a) show that also higher ACI values may contain information on aerosol (e.g. ACI<8). The MIPAS$-$*CALIOP* top

[Figure]

Figure 1: Relations between FOV along the line-of sight filled with cloud, extinction, and detected cloud top altitude. a) FOV area integrated along the line-of sight filled with cloud as a function of tangent altitude. b) FOV volume integrated along the line-of sight at detected cloud top as a function of extinction. c) Aerosol optical volume (AOV) volume at detected cloud top as a function of extinction. d) Cloud top difference (detected − simulated) as a function of AOV. e) (1-exp(-AOV)) as a function of extinction. f) Cloud top difference as a function of (1-exp(-AOV)).

[Figure]

Figure 2: Sensitivity of higher ACI values towards aerosol. a) A selected MIPAS orbit showing the aged Nabro aerosol layer having ACI values higher than 7. b) Cloud top height difference between MIPAS and CALIOP for the Nabro aerosol using MIPAS ACI thresholds of 7, 8, and 9. The data points for ACI 8 and 9 are shifted on the x-axis for better visibility.

altitude difference for the ACI thresholds of 7, 8, and 9 is shown in Fig. 2b. A higher ACI threshold leads to higher MIPAS top altitudes. For an ACI threshold of 8 the spread of the data remains constant and within a realistic range (overestimation still can be explained by the width of the FOV), but for an ACI threshold of 9 the spread of the differences nearly doubled and there are many overestimating cases that cannot be explained by the FOV. Increasing the ACI threshold also means increasing the detection sensitivity. Based on previous studies (Griessbach et al., 2014, 2016) we did not expect MIPAS to be sensitive to extinctions below $10^{-4}$km$^{-1}$. To extend the lower sensitivity range new simulations are subject to ongoing (for PSCs) and future work. Further, to identify the ACI thresholds that clearly separate between clear air variability and aerosol/clouds a new study similar to Sembhi et al. (2012) would be helpful, yet is beyond the scope of this study.

*3) Is the MIPAS instrument noise included in the sensitivity study? Also what is the impact of any residual radiometric offset after the calibration? Presumably this would have some effect on detection sensitivity if it produced a non-zero offset. Perhaps the Kleinert report has something on this?*

Indeed, our manuscript does not include a discussion on the impact of the instrument errors on the cloud detection. In response, we added a subsection and a new Fig. 3 on this topic to Section 3 in the revised manuscript.

**"3.3 Error estimation"**

"The idealised simulations above do not contain any instrument errors. To assess the potential impact of MIPAS instrument errors on the cloud detection with the ACI, we compared the increase in radiance due to aerosol/clouds with the average noise equivalent spectral radiance (NESR) of $2 \times 10^{-4}$ W m$^{-2}$ sr$^{-1}$ cm for the optimised resolution mode, the total scaling accuracy of 2.4 %, and the total offset accuracy of $9.5 \times 10^{-5}$ W m$^{-2}$ sr$^{-1}$ cm for band A (Kleinert et al., 2018). Although by absolute value the NESR appears to be the largest error it is reduced by $\sqrt{n}$, because we averaged over $n = 17$, 34, and 129 spectral points in the three ACI windows respectively.

[Figure]

Figure 3: Increase in radiance (cloud − clear air simulation) due to different clouds (solid coloured lines) compared to NESR (black solid line), offset accuracy (black dashed line), and scaling accuracy (dotted coloured lines; very close envelopes around solid coloured lines) for sulphate aerosol in a) the CI window around $792\,cm^{-1}$ (CI1), b) the CI window around $833\,cm^{-1}$ (CI2), c) the AI window around $960\,cm^{-1}$, and d) for ice, e) and ash. The colours indicate the background atmosphere type: red - tropics, yellow - mid-latitudes, light blue - polar summer, dark blue - polar winter. Each line represents one particle size distribution. The black symbols indicate the detected cloud top altitude using ACI=7.

The exemplary selected profiles for the three windows in Fig. 3 show that the increase in radiance at cloud top altitude is well above the NESR and the offset error, while the scaling accuracy results in a very tight envelope around the increase in radiance. From this consideration we deduced that using an ACI value above 7 ensures that no instrument effects are accidentally interpreted as a cloud. Moreover, Fig. 3 shows that the particle size distribution causes a significant spread of the increase in radiance (for ice clouds, sulphate aerosol ,and ash with $\beta_e = 1 \times 10^{-3}\,km^{-1}$ the spread at cloud top is $6.9 \times 10^{-3}$, $7.5 \times 10^{-4}$, and $1.1 \times 10^{-3}\,W\,m^{-2}\,sr^{-1}$ cm, respectively), and hence can be considered an important source of variability."

**Minor Comments**

*P1 L15: "In the fresh to two months ...". I would suggest rephrasing as "For plumes up to two months old ..."*
Done.

*P2 L50: I am surprised that there is no mention of ACE/SCISAT among the solar occulation instruments. However, perhaps they just focus on deriving molecular concentrations.*

We did not aim for a complete list of solar occultation instruments. However, considering your comment, we came to the conclusion to add ACE/SCISAT, because it is still operational and aerosol properties can be retrieved from all three instruments of

SCISAT: MAESTRO (McElroy et al., 2007), ACE-FTS (Doeringer et al., 2012), and the NIR imager (Vanhellemont et al., 2008).

We added the two most used SCISAT instruments: "... e.g. Stratospheric Aerosol and Gas Experiments II, Halogen Occultation Experiment, Polar Ozone and Aerosol Measurement (SAGE, HALOE, POAM, respectively, Randall et al., 2001), Atmospheric Chemistry Experiment Fourier Transform Spectrometer (ACE-FTS), ACE imager (Doeringer et al., 2012; Vanhellemont et al., 2008, ,respectively),..."

*P4 L96: 'index cloud detection' - should this be 'cloud index detection'?*
We changed it to "... MIPAS aerosol and cloud index methods ..."
*P4 L119: Mention Envisat's sun-synchronous orbit so that the concent of a fixed mean local solar time makes sense.*
We added the sun-synchronous orbit:
"As Envisat flew in a sun-synchronous orbit MIPAS measured at day- and nighttime with a mean local solar time of around 22:00 for the ascending node and 10:00 for the descending node."

*P5 L125-135: Presumably the pairs of number in brackets refer to the wavenumber range rather than a pair of spectral points, so I'd suggest using '–' (latex) rather than a comma. It would be helpful to have a diagram showing the spectra signatures being detected using these windows. Also, these all lie within the band A, so how is band B used?*

We now use a "–" (en-dash) to indicate the wavenumber range. Band B is used for the discrimination between aerosol and clouds. We added the details on the window ranges used for the aerosol ice discrimination:
"In the second step the discrimination between aerosol and ice clouds is performed using threshold functions for brightness temperature difference correlations between the following windows: $830.6-831.1\,\mathrm{cm}^{-1}$, $960.0-961.0\,\mathrm{cm}^{-1}$, and $1224.1-1224.7\,\mathrm{cm}^{-1}$."
and the ash filtering:
"In order to ensure that the MIPAS aerosol measurements used in our study only comprise sulphate aerosol, first the aerosol and cloud detection was performed, the ice clouds were filtered out as described above, and finally the volcanic ash detection method using two additional windows at $825.6-826.3\,\mathrm{cm}^{-1}$ and $950.1-950.9\,\mathrm{cm}^{-1}$ (Griessbach et al., 2014), which was also found to be sensitive to mineral dust and wild fire aerosol, was applied to filter out other non-sulphate UTLS aerosol types."
The different spectral signatures of the extinction coefficients covering band A and B are already shown in Fig. 3 in Griessbach et al. (2016). To give an impression of the impact on the radiance spectra we added a new Figure 1 to the revised manuscript (here, Fig.4) and the following description:
"The characteristic spectral signatures of clear air, an ice cloud, and a sulfate aerosol layer are illustrated in Fig 1 with selected measured MIPAS spectra, showing the broadband signatures in band A and B with a $1\,\mathrm{cm}^{-1}$ smoothed spectrum (Fig. 1a) and the relevant window regions with the original high spectral resolution (Fig. 1b)."

*P5 L136: 'that has flown ... since' rather than 'that flies ... since'.* Done.
*P5 L149: Either 'between ... and ...' or 'from ... to ...'* Changed to 'between ... and ...'.
*P6 L166: 'on average' rather than 'in average'* Done.
*P7 L189: Suggest 'where, potentially, cirrus clouds ...'* Done.
*P7 L205: If using 'the' for dates, then 'the 4th and the 25th'* Removed "the".

[Figure]

[Figure]

Figure 4: Representative measured MIPAS spectra for clear air, ice cloud, and aerosol measured on 14 June 2011 between 17 – 18 km altitude. a) MIPAS band A and B spectra smoothed to $1\,\mathrm{cm}^{-1}$ spectral resolution. b) High resolution MIPAS spectra only in the window regions used for aerosol/cloud detection and discrimination. The grey background indicates the window regions.

*P8 L224: Either 'These ... are ...' or 'This ... is ...'* Changed to "These ... are ...".

*P9 L268: Comparing Fig 1b and Table 2 it looks as if Fig 1b shows the differences after transferring to the coarser MIPAS sampling, whereas the text seems to imply Fig 1b is before the transfer.*

The coloured symbols in Fig. 2b show the differences for each scenario on the fine grid as given in the middle column of Table 2. Transferring each scenario to the coarser MIPAS sampling and assuming the worst case resulted in the grey symbols in Fig. 2b, which is the lower value of the range given in the right column of Tab. 2. To make this clearer, we now write:

"The fine grid simulation results show that the cloud top height mainly depends on the extinction coefficient, some effect can be seen for the background atmosphere, and nearly no effect can be seen for the different particle types (Fig. 2b, coloured symbols)."
and
"The lowermost top height (Fig. 2b, grey symbols) ..."

*P14 L445: 'discrepancy'* Done.
*P18 L562: 'contradictory'* Done.

*Table 2: hard to distinguish between '-' indicating a range of values and a negative sign. In LaTeX use either '–' or '−'.*
We use L\&TeX. For the minus sign (−) we used: "$-$" and for the range (–) we used the en-dash "--", which, to our knowledge, is in line with the conventions. Should discrepancies on this point remain we would like to leave this to the Copernicus copy editing (conventions).

*Fig 7: Not clear what CALIOP aerosol maximum extinction means. If this is the maximum value of extinction in the CALIOP profile (equivalent to that found by the subsequent gradient method) then that may be lower in altitude than the part of the profile which MIPAS is detecting. So perhaps something else is meant by 'maximum'?*

Fig. 7 (Fig. 9 in the revised manuscript) was made using an extinction coefficient threshold of $3 \times 10^{-3} \, \text{km}^{-1}$ to derive the aerosol layer top altitude, which is also stated in the text (P13 L385 in the revised manuscript) in the paragraph before Fig. 9 is discussed. To find out if the top altitude difference between MIPAS and CALIOP depends on the aerosol layer optical thickness, Fig. 9a indeed shows the difference (still using $3 \times 10^{-3} \, \text{km}^{-1}$ to derive top altitude), but as a function of the maximum extinction coefficient of that aerosol layer. Actually, the average extinction or the AOD would have been a better measure, but since the CALIOP aerosol profiles terminate between $12 - 14 \, \text{km}$ we decided to do this analysis with the maximum extinction value of each layer. For clarification we added to the figure caption:

"Cloud top height differences between MIPAS and CALIOP Nabro aerosol measurements (using a CALIOP extinction threshold of $3 \times 10^{-3} \, \text{km}^{-1}$) as a function of a) CALIOP aerosol layer maximum extinction, ..."
and
"c) shows the diagnostic relation between CALIOP maximum extinction and MIPAS minimum ACI."

*Fig 8 caption: on the right hand plots, what are the grey boxes and beta numbers?*
We added: "The grey boxes indicate the cloud layers detected by the lidar and $\beta_e$ gives the average layer extinction."
*Fig 9 caption: 'Only for the twilight measurements is there ...'*
Done.
*Fig 13: given the standard representation is to scale aerosol extinctions to 550 or 532nm, I think it would be useful to add a second x-axis along the top with this scaling as well.*
We agree and added a second x-axis.

**References**

Doeringer, D., Eldering, A., Boone, C. D., González Abad, G., and Bernath, P. F.: Observation of sulfate aerosols and SO2 from the Sarychev volcanic eruption using data from the Atmospheric Chemistry Experiment (ACE), J. Geophys. Res., 117, https://doi.org/10.1029/2011JD016556, URL `https://agupubs.onlinelibrary.wiley.com/doi/abs/10.1029/2011JD016556`, 2012.

Griessbach, S., Hoffmann, L., Spang, R., and Riese, M.: Volcanic ash detection with infrared limb sounding: MIPAS observations and radiative transfer simulations, Atmos. Meas. Tech., 7, 1487–1507, https://doi.org/10.5194/amt-7-1487-214, 2014.

Griessbach, S., Hoffmann, L., Spang, R., von Hobe, M., Müller, R., and Riese, M.: Infrared limb emission measurements of aerosol in the troposphere and stratosphere, Atmos. Meas. Tech., 9, 4399–4423, https://doi.org/10.5194/amt-9-4399-2016, URL `http://www.atmos-meas-tech.net/9/4399/2016/`, 2016.

Kleinert, A., Birk, M., Perron, G., and Wagner, G.: Level 1b error budget for MIPAS on ENVISAT, Atmos. Meas. Tech., 11, 5657–5672, https://doi.org/10.5194/amt-11-5657-2018, URL `https://www.atmos-meas-tech.net/11/5657/2018/`, 2018.

McElroy, C. T., Nowlan, C. R., Drummond, J. R., Bernath, P. F., Barton, D. V., Dufour, D. G., Midwinter, C., Hall, R. B., Ogyu, A., Ullberg, A., Wardle, D. I., Kar,

J., Zou, J., Nichitiu, F., Boone, C. D., Walker, K. A., and Rowlands, N.: The ACE-MAESTRO instrument on SCISAT: description, performance, and preliminary results, Appl. Optics, 46, 4341–4356, https://doi.org/10.1364/AO.46.004341, URL `http://ao.osa.org/abstract.cfm?URI=ao-46-20-4341`, 2007.

Sembhi, H., Remedios, J., Trent, T., Moore, D. P., Spang, R., Massie, S., and Vernier, J. P.: MIPAS detection of cloud and aerosol particle occurrence in the UTLS with comparison to HIRDLS and CALIOP, Atmos. Meas. Tech., 5, 2537–2553, https://doi.org/10.5194/amt-5-2537-2012, 2012.

Vanhellemont, F., Tetard, C., Bourassa, A., Fromm, M., Dodion, J., Fussen, D., Brogniez, C., Degenstein, D., Gilbert, K. L., Turnbull, D. N., Bernath, P., Boone, C., and Walker, K. A.: Aerosol extinction profiles at 525 nm and 1020 nm derived from ACE imager data: comparisons with GOMOS, SAGE II, SAGE III, POAM III, and OSIRIS, Atmos. Chem. Phys., 8, 2027–2037, https://doi.org/10.5194/acp-8-2027-2008, URL `https://www.atmos-chem-phys.net/8/2027/2008/`, 2008.

---

## Author Comment (AC2) · 6 Dec 2019

**Reply to Reviewer 2**

We thank the reviewer for reviewing our manuscript and providing comments, questions, food for thoughts, and corrections. We went through all points and provide answers to each of them. Where necessary, we also modified the manuscript. Please see below for the details.

**Specific comments:**

*MIPAS-simulations: To be able to estimate how much the instrumental performance itself contributes to the cloud/aerosol top height estimation, it would be interesting to estimate/discuss the errors introduced by the random and systematic uncertainties of MIPAS, e.g. spectral noise, radiometric accuracy, and others as described by Kleinert et al., 2018.*

Indeed, our manuscript did not include a discussion on the impact of the instrument errors on the cloud detection. As we have not presented our considerations in the submitted version, we added a subsection 3.3 and a new Fig. 3 on this topic to Section 3 in the revised manuscript.

**"3.3 Error estimation"**

"The idealised simulations above do not contain any instrument errors. To assess the potential impact of MIPAS instrument errors on the cloud detection with the ACI, we compared the increase in radiance due to aerosol/clouds with the average noise equivalent spectral radiance (NESR) of $2 \times 10^{-4}\,\mathrm{W\ m^{-2}\ sr^{-1}}$ cm for the optimised resolution mode, the total scaling accuracy of 2.4 %, and the total offset accuracy of $9.5 \times 10^{-5}\,\mathrm{W\ m^{-2}\ sr^{-1}}$ cm for band A (Kleinert et al., 2018). Although by absolute value the NESR appears to be the largest error it is reduced by $\sqrt{n}$, because we averaged over $n =$ 17, 34, and 129 spectral points in the three ACI windows respectively. The exemplary selected profiles for the three windows in Fig. 1 show that the increase in radiance at cloud top altitude is well above the NESR and the offset error, while the scaling accuracy results in a very tight envelope around the increase in radiance. From this consideration we deduced that using an ACI value above 7 ensures that no instrument effects are accidentally interpreted as a cloud. Moreover, Fig. 1 shows that the particle size distribution causes a significant spread of the increase in radiance (for ice clouds, sulphate aerosol ,and ash with $\beta_e =1 \times 10^{-3}\,\mathrm{km^{-1}}$ the spread at cloud top is $6.9 \times 10^{-3}$, $7.5 \times 10^{-4}$, and $1.1 \times 10^{-3}\,\mathrm{W\ m^{-2}\ sr^{-1}}$ cm, respectively), and hence can be considered an important source of variability."

*Could you discuss whether it would make sense with respect to detection sensitivity, to use the radiances at the maximum of the sulfate peak around 1100 cm-1 instead of those around 800 cm-1 for the aerosol detection since the absorption seems to be an order of magnitude higher?*

The main reason for not selecting a window in band AB $(1020 - 1170\,\mathrm{cm^{-1}})$ is that we did not find a useful window there. Comparing the radiance contour plot of a non-cloudy profile for band A (Fig. 2a) with band AB (Fig. 2b), the window regions around $832\,\mathrm{cm^{-1}}$ and $960\,\mathrm{cm^{-1}}$ clearly stand out with low radiances (dark colours) even at

[Figure]

Figure 1: Increase in radiance (cloud − clear air simulation) due to different clouds (solid coloured lines) compared to NESR (black solid line), offset accuracy (black dashed line), and scaling accuracy (dotted coloured lines; very close envelopes around solid coloured lines) for sulphate aerosol in a) the CI window around $792\,cm^{-1}$ (CI1), b) the CI window around $833\,cm^{-1}$ (CI2), c) the AI window around $960\,cm^{-1}$, and d) for ice, e) and ash. The colours indicate the background atmosphere type: red - tropics, yellow - mid-latitudes, light blue - polar summer, dark blue - polar winter. Each line represents one particle size distribution. The black symbols indicate the detected cloud top altitude using ACI=7.

the lowest altitudes. In contrast, in band AB there is no obvious window region. In the entire band AB $O_3$ has a strong impact, which actually impairs relatively weak aerosol signals at altitudes below the ozone layer. Further, there are contributions by other species, such as $CO_2$, CFC-11, CFC-12, HCFC-22, $N_2O$, that also interfere with UTLS aerosol signals.

[Figure]

Figure 2: Radiances of a clear air profile in a) band A and b) band B.

*Regarding the comparison with CALIOP: could the variability of the CALIOP aerosol top height within the match-criteria be used to estimate the plume's homogeneity at its upper level and be correlated with the MIPAS cloud-top in order to distinguish between cloud-inhomogeneity and optical thickness as the reason for the underestimation by MIPAS?*

In our study the CALIOP data are already averaged over 1°, corresponding to 111 km, to achieve a sufficient detection sensitivity for the comparison with MIPAS. Hence, a lot of variability in the CALIOP data is already smoothed out. Fig. 3a shows that the standard deviation decreases over time, which can be expected for an aging aerosol plume. The top altitude difference shows only a slight dependency on the standard deviation of the CALIOP aerosol top height (Fig. 3b) and thus cannot be used to distinguish between cloud-inhomogeneity and optical thickness as the reason for the underestimation by MIPAS.

[Figure]

Figure 3: Standard deviation $\sigma$ of the CALIOP aerosol layer top height within the MIPAS match radius a) Evolution of $\sigma$ with time b) top altitude difference between MIPAS and CALIOP aerosol detections as a function of $\sigma$.

*Throughout the paper it is argued with extinction. However, would a quantity like optical depth covered by the field-of-view not be better suited?*

Yes, we thought so too and investigated if a 3D equivalent to AOD (path length in cloud $\times$ extinction), the aerosol optical volume (*AOV = integrated FOV along the line of sight in cloud $\times$ extinction*[3]) would be a better parameter. We started with a 2D setup and integrated the vertically oriented area filled with cloud for each tangent point (Fig 4a). In the following we used only the tangent heights of the detected cloud top heights. To extend the area to a (3D) volume, we assumed the 3D FOV as a rectangular tube and multiplied the area by 30 km. Fig 2a in the revised manuscript and Fig. 4b in the reply show the integrated FOV-volume along the LOS at the detected cloud top as a function of cloud extinction. Analogously to AOD we then calculated the AOV. In contrast to the expectation that the AOV at cloud detection should be around a constant value, the AOV at the detected cloud top shows an exponential dependency on extinction (Fig. 4c). Using AOV instead of extinction in Fig. 4d gives a very similar picture to Fig. 1b in the manuscript. Since the extinction is the dominant factor and allows for a better comparison with the full suite of other instruments, we prefer presenting the results as a function of extinction.

*L534-542: It should be made clear that these considerations are valid for the typical size distribution of sulfate aerosols. Could you also consider/discuss cases for other particle sizes (e.g. smaller particles) where scattering in the UV/VIS is decreased but the absorption signal in the mid-IR is not/less affected?*

To make it clearer that these considerations are only valid for sulfate aerosol, we added a scaling curve for stratospheric background aerosol at 20 km (Deshler et al., 2003,

[Figure]

Figure 4: Extinction and field of view diagnostics. a) FOV area filled with cloud as a function of tangent altitude for the simulation scenarios. The grey region indicate the vertical cloud extent. The coloured lines (atmosphere) give the area in cloud for each tangent altitude. b) FOV volume integrated along the line of sight in cloud. Same as Fig. 1a in the manuscript. c) Aerosol optical volume (AOV) at detected cloud top altitude as a function of extinction. d) Cloud top altitude difference as a function of AOV.

($r_{\text{eff}} = 0.13\,\mu$m)), which is very similar to the "Nabro" scaling curve, to Fig. 4 in the revised manuscript and we added to the text:
"... we used the scaling factors for background aerosol and volcanically enhanced sulphate aerosol with particle sizes larger than 0.1 $\mu$m from Fig. 2 and Table 3 to compare the sensitivity range of MIPAS ..."
and to the figure caption:
"... and background aerosol at 20 km (black solid line Deshler et al., 2003, April 1999)."
For smaller particles the scaling factor from mid-IR to UV/VIS would be lower. However, we did not consider smaller particle sizes than the Deshler measurements indicate for the background state. In Griessbach et al. (2014, Tab. 5) we found that for particle size distributions with effective radii smaller than 100 nm unrealistically high number concentrations are required in order to achieve extinctions that are detectable by MI-PAS. Also, for SAGE II Thomason et al. (2008) found a "lack of sensitivity to particles with radii less than 100 nm". Since the satellite measurements compared here do not have sufficient sensitivity to these smaller particles, which no doubt exist in the UTLS (e.g. Clarke and Kapustin, 2002), we refrained from adding this discussion to the paper.

*Table 1: SCIAMACHY and OMPS NPP may be added. The first one since it could be*

*directly compared to MIPAS in future work and the second to cover the present time and the future.*

Thanks for the suggestion. We initially started a more comprehensive Tab. 1 including way more instruments. But as this study is not meant as a satellite aerosol remote sensing review, we decided to include only one representative instrument for each measurement principle. The aerosol measurements of SCIAMACHY and OMPS/LP both rely on the solar scattering technique as OSIRIS, which is still measuring. Studies on OMPS/LP by e.g. Jaross et al. (2014); Flynn et al. (2007) point out the similarities to OSIRIS and SCIAMACHY. Please find below the values we compiled for SCIA-MACHY and OMPS/LP. As you will notice, for SCIAMACHY and OOMPS/LP we could not find out all values for the parameters we are listing in Tab. 1 in the manuscript. We finally selected OSIRIS, because we could not find conclusive values (on the sensitivity range in particular) for SCIAMACHY and OMPS/LP in literature.

Table 1: Overview of relevant instrument characteristics for global aerosol measurements that also rely on the solar scattering technique as OSIRIS, which we selected as a representative in the manuscript.

| instrument | channel | sensitivity range | vertical sampling | coverage | profiles per day | comments & references |
|---|---|---|---|---|---|---|
| OMPS/LP | 675 nm[1] | min:
max: | 1 km [2] | 82°N – 82°S [1] | $\sim$ 7000 [2] | **daytime** extinction profiles
[1]Chen et al. (2018); extinction range in figures: $1 \times 10^{-5}$ – $1 \times 10^{-2}$ km$^{-1}$
[2]Rault and Loughman (2013) |
| SCIAMACHY | 750 nm [3] | min:
max: | 3.3 km [3] | $\sim$82°N – 82°S [4]
winter hemisphere[5]:
50 – 60°S;
60 – 70°N | 320 – 430 [6,7] | **daytime** extinction profiles
[3,4,5,6,7](Rieger et al., 2018; Bovensmann et al., 1999; Weigel et al., 2016; Kaiser et al., 2004; Cardaci, 2010)
upper aerosol limits (cloud filtering)
extinction range in figures: $1 \times 10^{-5}$ – $1 \times 10^{-2}$ km$^{-1}$
(Taha et al., 2011) |

**Technical comments:**

*L50: 'occulation' − > 'occultation'* Done.
*L69: why is 'However' used here?* Removed.
*L125: 'color' vs. Figure 1 caption: 'colours', please harmonize* Done.
*L269: 'maximal' − > 'maximum'* Done.
*L306: 'compareable' − > 'comparable'* Done.
*L416: delete ')'* Done.
*L417: 'analysed' but also 'analyzed' is used* Done.
*L445: 'dicrepancy' − > 'discrepancy'* Done.
*L450: 'exinction' − > 'extinction'* Done.
*L562: 'contradicory' − > 'contradictory'* Done.
*L659: 'underestmated' − > 'underestimated'* Done.
*L679: 'soon be available': is the dataset already available?*
It is now online (since 6 November 2019).
*Table1, last column: '1.5 km^-4 sr^-1' − > '1.5^-4 km^-1 sr^-1'* Fixed.
*Fig. 5, caption: 'inidcated' − > 'indicated'* Done.

*Fig. 9: could you also show a further panel with the absolute plume altitudes to better judge the difference compared to the absolute value. Fig. 9: a legend, e.g. in one of the panels indicating the different instruments would be better than only having the information in the caption.*

We added a panel to Fig. 9 (Fig. 11 in the revised manuscript) showing the absolute plume tops measured by MIPAS and the plume altitude (from top to bottom) measured by the lidar and twilight measurements. To this panel we also added the instrument names/ground station names in the corresponding colours. In addition we changed the figure caption to: "Properties of the Nabro sulphate aerosol as a function of time derived from MIPAS (light green), CALIOP (grey), Leipzig lidar (light blue), Jülich lidar (dark blue), Esrange lidar (black), and twilight measurements (dark green). ... a) Nabro aerosol layer height measured by all instruments. For MIPAS only the top height is shown. In case of multiple matches for MIPAS the minimum, maximum, and mean top heights derived are shown. ...", and also refer to the new panel a from the text, where appropriate: "In the comparison of the top heights (Fig. 9a) we see a decrease with time and when moving from low (CALIOP and twilight) to high latitudes (ground based lidars). Compared to the Leipzig lidar measurements ...".

*Fig. 13, caption: 'gray' − > 'grey'* Done.

**References**

Bovensmann, H., Burrows, J. P., Buchwitz, M., Frerick, J., Noël, S., Rozanov, V. V., Chance, K. V., and Goede, A. P. H.: SCIAMACHY: Mission Objectives and Measurement Modes, J. Atmos. Sci., 56, 127–150, https://doi.org/10.1175/1520-0469(1999)056$\langle$0127:SMOAMM$\rangle$2.0.CO;2, URL https://doi.org/10.1175/1520-0469(1999)056<0127:SMOAMM>2.0.CO;2, 1999.

Cardaci, M.: ENVISAT-1 Products Specifications, Tech. rep., ESA-ESRIN, URL https://earth.esa.int/c/document_library/get_file?folderId=13020&name=DLFE-621.pdf, 2010.

Chen, Z., Bhartia, P. K., Loughman, R., Colarco, P., and DeLand, M.: Improvement of stratospheric aerosol extinction retrieval from OMPS/LP using a new aerosol model, Atmos. Meas. Tech., 11, 6495–6509, https://doi.org/10.5194/amt-11-6495-2018, URL https://www.atmos-meas-tech.net/11/6495/2018/, 2018.

Clarke, A. D. and Kapustin, V. N.: A pacific aerosol survey. Part I: A decade of data on particle production, transport, evolution, and mixing in the troposphere, J. Atmos. Sci., 59, 363–382, 2002.

Deshler, T., Larsen, N., Weissner, C., Schreiner, J., Mauersberger, K., Cairo, F., Adriani, A., Di Donfrancesco, G., Ovarlez, J., Ovarlez, H., Blum, U., Fricke, K. H., and Dornbrack, A.: Large nitric acid particles at the top of an Arctic stratospheric cloud, J. Geophys. Res., 108, https://doi.org/10.1029/2003JD003479, 2003.

Flynn, L. E., Seftor, C. J., Larsen, J. C., , and Xu, P.: The Ozone Mapping and Profiler Suite, in: Earth Science Satellite Remote Sensing, edited by Qu, J. J., Gao, W., Kafatos, M., Murphy, R. E., and Salomonson, V. V., p. 279–296, Springer, Berlin, URL https://doi.org/10.1007/978-3-540-37293-6, 2007.

Griessbach, S., Hoffmann, L., Spang, R., and Riese, M.: Volcanic ash detection with infrared limb sounding: MIPAS observations and radiative transfer simulations, Atmos. Meas. Tech., 7, 1487–1507, https://doi.org/10.5194/amt-7-1487-214, 2014.

Jaross, G., Bhartia, P. K., Chen, G., Kowitt, M., Haken, M., Chen, Z., Xu, P., Warner, J., and Kelly, T.: OMPS Limb Profiler instrument performance assessment, J. Geophys. Res., 119, 4399–4412, https://doi.org/10.1002/2013JD020482, URL https://agupubs.onlinelibrary.wiley.com/doi/abs/10.1002/2013JD020482, 2014.

Kaiser, J., Eichmann, K.-U., Noel, S., Bovensmann, H., Burrows, J., and Frerick, J.: Satellite-pointing retrieval from atmospheric limb-scattering of solar UV-B radiation, Can. J. Phys., 82, 1041–1052, https://doi.org/10.1139/p04-071, 2004.

Kleinert, A., Birk, M., Perron, G., and Wagner, G.: Level 1b error budget for MIPAS on ENVISAT, Atmos. Meas. Tech., 11, 5657–5672, https://doi.org/10.5194/amt-11-5657-2018, URL https://www.atmos-meas-tech.net/11/5657/2018/, 2018.

Rault, D. F. and Loughman, R. P.: The OMPS Limb Profiler Environmental Data Record Algorithm Theoretical Basis Document and Expected Performance, IEEE Trans. Geosci. Remote Sens., 51, 2505–2527, https://doi.org/10.1109/TGRS.2012.2213093, 2013.

Rieger, L. A., Malinina, E. P., Rozanov, A. V., Burrows, J. P., Bourassa, A. E., and Degenstein, D. A.: A study of the approaches used to retrieve aerosol extinction, as applied to limb observations made by OSIRIS and SCIAMACHY, Atmos. Meas. Tech., 11, 3433–3445, https://doi.org/10.5194/amt-11-3433-2018, URL https://www.atmos-meas-tech.net/11/3433/2018/, 2018.

Taha, G., Rault, D. F., Loughman, R. P., Bourassa, A. E., and von Savigny, C.: SCIAMACHY stratospheric aerosol extinction profile retrieval using the OMPS/LP algorithm, Atmos. Meas. Tech., 4, 547–556, https://doi.org/10.5194/amt-4-547-2011, URL https://www.atmos-meas-tech.net/4/547/2011/, 2011.

Thomason, L. W., Burton, S. P., Luo, B.-P., and Peter, T.: SAGE II measurements of stratospheric aerosol properties at non-volcanic levels, Atmos. Chem. Phys., 8, 983–995, https://doi.org/10.5194/acp-8-983-2008, URL `https://www.atmos-chem-phys.net/8/983/2008/`, 2008.

Weigel, K., Rozanov, A., Azam, F., Bramstedt, K., Damadeo, R., Eichmann, K.-U., Gebhardt, C., Hurst, D., Kraemer, M., Lossow, S., Read, W., Spelten, N., Stiller, G. P., Walker, K. A., Weber, M., Bovensmann, H., and Burrows, J. P.: UTLS water vapour from SCIAMACHY limb measurements V3.01 (2002–2012), Atmos. Meas. Tech., 9, 133–158, https://doi.org/10.5194/amt-9-133-2016, URL `https://www.atmos-meas-tech.net/9/133/2016/`, 2016.